# Label-free neuroimaging in vivo using synchronous angular scanning microscopy with single-scattering accumulation algorithm

Moonseok Kim[1,2,3,4,8], Yonghyeon Jo [1,2,8], Jin Hee Hong[1,2], Suhyun Kim[5], Seokchan Yoon [1,2], Kyung-Deok Song[1,2], Sungsam Kang [6], Byunghak Lee [7], Guang Hoon Kim[7], Hae-Chul Park [5] & Wonshik Choi[1,2]

Label-free in vivo imaging is crucial for elucidating the underlying mechanisms of many important biological systems in their most native states. However, the applicability of existing modalities has been limited to either superficial layers or early developmental stages due to tissue turbidity. Here, we report a synchronous angular scanning microscope for the rapid interferometric recording of the time-gated reflection matrix, which is a unique matrix characterizing full light-specimen interaction. By applying single scattering accumulation algorithm to the recorded matrix, we removed both high-order sample-induced aberrations and multiple scattering noise with the effective aberration correction speed of 10,000 modes/s. We demonstrated in vivo imaging of whole neural network throughout the hind-brain of the larval zebrafish at a matured stage where physical dissection used to be required for conventional imaging. Our method will expand the scope of applications for optical imaging, where fully non-invasive interrogation of living specimens is critical.

[1] Center for Molecular Spectroscopy and Dynamics, Institute for Basic Science, Seoul 02841, Korea. [2] Department of Physics, Korea University, Seoul 02841, Korea. [3] Department of Medical Life Sciences, College of Medicine, The Catholic University of Korea, Seoul 06591, Korea. [4] Department of Biomedicine & Health Sciences, The Catholic University of Korea, Seoul 06591, Korea. [5] Department of Biomedical Sciences, Korea University, Ansan 425-707, Korea. [6] Laser Biomedical Research Center, G. R. Harrison Spectroscopy Laboratory, Massachusetts Institute of Technology, Cambridge, MA 02139, USA. [7] Korea Electrotechnology Research Institute, Ansan 15588, Korea. [8] These authors contributed equally: Moonseok Kim, Yonghyeon Jo. Correspondence and requests for materials should be addressed to W.C. (email: wonshik@korea.ac.kr)

Light waves propagating in biological tissues experience wavefront distortion due to the complex spatial heterogeneity of the refractive index. The wavefront distortion, excluding the severe multiple-scattering noise, is often called a sample-induced aberration, which manifests itself as the phase retardation of propagating waves depending on their propagation angles. Considering that a focused spot is formed by the coherent superposition of many angular waves, it is evident that this sample-induced aberration causes the improper formation of a focus, thereby reducing both the resolving power and signal to the multiple-scattering noise ratio. In recent years, various methods have been proposed to deterministically make use of multiple-scattered waves[1–7]; however, the prevailing methods for biological imaging mainly involve compensating the sample-induced aberration in the context of adaptive optics (AO)[8,9]. Typical AO microscopy for deep-tissue imaging has been designed to work for fluorescence imaging, and it has played a pivotal role in elucidating the underlying mechanisms of biological systems that could not otherwise be visualized. It identifies the sample-induced aberration by either direct wavefront sensing[10–12] or feedback control of wavefront-shaping devices[13–16]. However, in both cases, it is necessary to iteratively update the aberration correction to enhance the signal of interest, which is initially lower than the background fluorescence noise induced by multiple scattering in the case of deep-tissue imaging. Owing to the incoherent nature of fluorescence emission, this corrective update can only be made by multiple image acquisitions at each iteration step, which slows down the overall image acquisition speed. It typically takes more than 10 s to complete the entire sequence of aberration correction, during which the sample must be stationary. An important trend in recent years has been to mitigate the speed limitation by simultaneously correcting multiple sub-pupils[17] or sub-areas[18] in the large view field. This has effectively increased the image acquisition speed in terms of the corrected area per unit time. Equivalently, the number of corrected modes per unit time has been increased to approximately 50 modes/s, but the minimum image acquisition time has remained the same.

AO has also been applied to coherent imaging modalities such as confocal reflectance imaging, optical coherence tomography, and digital holographic microscopy for the applications where administration of labeling agents needs special care. For instance, individual photoreceptor cells in the retina of a living eye were clearly visualized by sensing and correcting the ocular aberrations[19–23]. Notably, software-based AO microscopy has been developed, where aberration correction is applied after the image acquisition[24–27]. This is available only for the coherent imaging modalities because the phase of the scattered waves is recorded. These software-based AO imaging methods can be much faster than the hardware-based AO imaging because sample needs to be stationary only during the one-time image acquisition. Furthermore, the cost of building a system can be reduced owing to the exemption of the wavefront-shaping devices. However, most of the developed algorithms require point-like structures that can serve as guiding stars because of their inability to distinguish the aberrations on the way to the specimens from those on the way out. Alternatively, an illumination beam with narrow angular divergence was used to minimize the aberration on the way in, and image metrics such as image sharpness and intensity were optimized by adding corrective phases to the Fourier-transformed map of the acquired image[28–30]. Because these approaches cannot easily distinguish signals from multiple-scattering noise, they are susceptible to the multiple-scattering noise. For this reason, the applications of AO coherent imaging have been confined to weak-scattering cases such as retinal imaging. To overcome strong multiple scattering noise and sample-induced aberration, time-gated reflection matrix approaches[31,32] have been investigated,

which is composed of a set of wide-field and time-gated complex field maps of intrinsic elastic backscattering taken for various illumination angles. However, this approach has been inapplicable for the in vivo imaging because a slow liquid-crystal spatial light modulator should be used for the recording of the time-gated reflection matrix.

Here we develop an adaptive optical synchronous angular scanning microscope (AO-SASM) for in vivo deep-tissue imaging free from any labeling agents. In AO-SASM, we realized the high-speed recording of a time-gated reflection matrix by synchronously scanning the angle of the sample and reference waves from a supercontinuum laser. This strategy ensures the uniform interference over the wide view field even when both temporal pulse front and wavefront are rotated by fast scanning mirrors. By processing this matrix in such a way to coherently accumulate single scattering signal, we removed both multiple scattering noise and local high-order sample-induced aberration. In doing so, we significantly reduced the image acquisition time per depth from a few minutes to 0.22 s and enhanced the effective aberration correction speed excluding the data processing time to 10,000 modes/s (see Supplementary Note 6). With the high-speed imaging capability of AO-SASM, we performed in vivo volumetric imaging of a living zebrafish as old as 21 days post-fertilization (dpf) and visualized the fine axon branches of reticulospinal neurons and torus semi-circularis in the entire neural network by the ideal diffraction-limit spatial resolution (370–480 nm). The identified nanostructures are responsible for the important biological functions such as fast turning movements and transfer auditory stimuli. In the conventional imaging modalities such as confocal fluorescence/reflectance microscopy, optical coherence microscopy and multi-photon microscopy, these structures have been inaccessible in the matured stage without the dissection of the specimens due to the severe sample-induced aberrations.

## Results

**Principle of adaptive optical synchronous angular scanning microscopy.** For the recording of a time-gated reflection matrix, the incidence angle, or equivalently the transverse wavevector $\mathbf{k}^i$, of a planar wave needs to be scanned for identifying the angle-dependent phase retardation[32]. However, this strategy generates a complication that does not exist in focus scanning microscopy, especially when it is combined with time-gated detection. In the focus scanning microscope, the focused spot remains stationary at the detector during the scanning because the backscattered wave retro-reflected at the sample can be de-scanned by the scanning mirrors. However, in the case of planar illumination, the temporal pulse front of the backscattered waves is tilted to the opposite angle with respect to the surface normal such that the de-scanning action does not occur at the scanning mirror. Therefore, the scanning of $\mathbf{k}^i$ gives rise to the rotation of the temporal pulse front at the detector plane in the case of time-gated detection. For interferometric detection of the backscattered waves with a reference wave whose wavefront and pulse front are fixed parallel to the detector plane, the interference occurs only in the narrow area within the field of view (Fig. 1b, d).

To resolve this issue, we previously used a liquid-crystal spatial light modulator (SLM) because it makes the pulse front of the modulated beam remain parallel to the surface of the SLM even with the scanning of $\mathbf{k}^i$. The drawback was the slow angular scanning rate (≤30 Hz) due to the long response time of the liquid crystal. Herein, we propose a method employing a pair of galvanometer scanning mirrors (GM) for the fast scanning of $\mathbf{k}^i$, while maintaining the interference over the full field of view in the time-gated imaging. In our proposed method, we scanned the

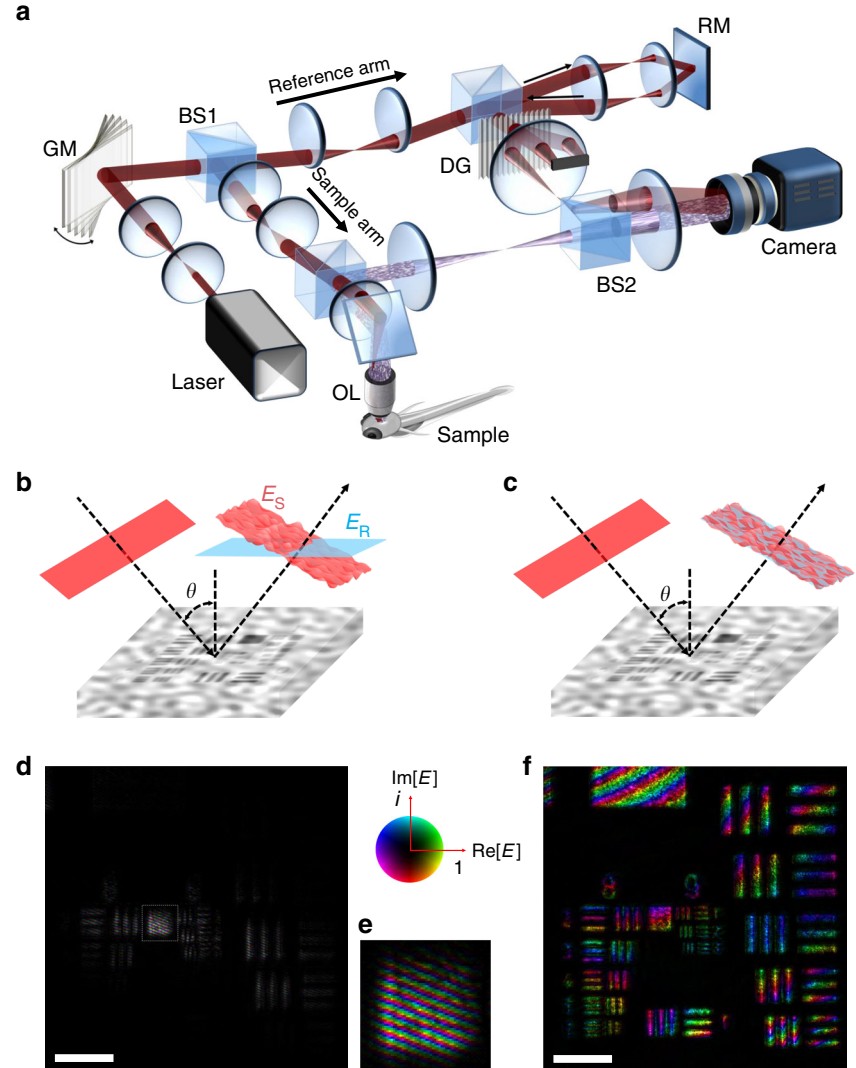

**Fig. 1** Schematic layout and working principle of AO-SASM. **a** Layout of the experimental setup. A broadband supercontinuum laser (NKT Photonics, model EXR-15) with a short temporal coherence time was used for the time-gated detection. The first-order diffraction from a diffraction grating was used as a reference wave. Camera: PCO; edge rolling shutter 4.2. For comparison with AO-SASM, the confocal reflectance and fluorescence microscopes were constructed using the same objective lens and sample stage. For the detailed experimental setup, see Supplementary Note 1. **b** Interference image in the case of a fixed reference wave. When the incidence angle is scanned by the GM, the pulse fronts of the sample and reference waves are mismatched. For visibility, the pulse front of the reference wave is colored in cyan, although its wavelength is the same as that of the sample wave. **c** Synchronous rotation of sample and reference waves. Interference occurs throughout the entire field of view. **d** Complex field map acquired for the case of a fixed reference with the oblique illumination. Scale bar, 20 μm. Color bar: The saturation and color of the color bar indicates the amplitude and phase of the complex field, respectively. **e** White dashed box in (**d**) is zoomed in, where fine fringes appear due to the angle difference between the sample and reference waves. **f** The complex field map acquired for the case of synchronous rotation of sample and reference waves with the oblique illumination. See Supplementary Note 2 for the detailed comparison between the fixed reference and the rotating reference. GM Galvanometer scanning mirror (Cambridge Technology 6220H) for scanning the angle of illumination to the sample and reference, BS1, 2 beam splitters for splitting and combining the sample and reference waves, RM reference mirror to adjust the gating time, OL objective lens; DG: diffraction grating

wavevector of the reference wave $\mathbf{k}_R^i$ in conjunction with $\mathbf{k}^i$ using the same scanning mirror. This was realized by installing the GM before dividing the sample and reference waves by a beam splitter (BS1) (Fig. 1a). The pulse fronts of both the sample and reference waves rotate synchronously, which leads to the uniform interference over the entire view field (Fig. 1c, f), especially for the single-scattered waves that we are interested in. Since images were taken in the rotating reference frame, a careful conversion of basis to the laboratory frame was devised in the image processing step. In doing so, AO-SASM increased the data acquisition speed (≥450 frames per second) by more than an order of magnitude compared with our previous study. In fact, the acquisition speed

was mainly limited by the camera speed, not by the beam scanning speed. With the minimization of the number of required images, the effective imaging speed was increased by another order of magnitude.

In the AO-SASM, we used a supercontinuum laser as a light source whose center wavelength and bandwidth are tunable. The center wavelength was chosen in the range between 450 and 650 nm depending on the spatial resolution of interest. The typical bandwidth used in the experiment was 15 nm. This corresponds to the temporal coherence time of ~100 fs, which is the effective width of the pulse front, and coherent gating of 15 μm. An objective lens with a numerical aperture (NA) of 0.8

was used to deliver the planar illumination to the sample and capture backscattered waves. Therefore, single-depth recording of complex-field images can cover a depth range of 15 μm by the axial resolution of 2.0 μm set by confocal gating with the use of the computational refocusing. A camera recorded the interferogram formed by the backscattered wave from the sample and a reference wave with a magnification of 120× (see "Methods" section for further details on the experimental setup). Angular scanning of the illumination uniformly covered the entire NA of the objective lens, and the number of illumination angles $N_{in}$ varied between 100 and 800 depending on the degree of aberration and multiple-scattering noise. $N_{in}$ can be significantly smaller than the number of free modes determined by the number of diffraction-limit spots in the view field, because each recorded image contains aberration information for all the modes (see Supplementary Notes 6 and 7 for optimal $N_{in}$). From one set of angle-scanned backscattered images taken for a fixed target depth, we constructed a time-gated reflection matrix $E_S(\mathbf{r}^o; \mathbf{r}^i, \tau_0)$, which describes the complex field map of backscattered wave at the image plane $\mathbf{r}^o$ for the illumination of a wave at a position $\mathbf{r}^i$. In this construction, the original images taken with respect to the rotating reference waves were converted to the images in the laboratory frame (see Methods for the construction of $E_S(\mathbf{r}^o; \mathbf{r}^i, \tau_0)$). Here, the flight time $\tau_0$ where the temporal gating was applied was set by the position of a reference mirror (RM). With this reflection matrix, we corrected the sample-induced aberration by improving the algorithm called as collective and closed-loop accumulation of single scattering[31,32], in which sample-induced aberrations are identified from the measured time-gated reflection matrix, preferably in such a way to maximize the single-scattered waves over multiple-scattering noise. Conventional confocal reflectance/fluorescence microscopy was set up on the top of the AO-SASM for the comparison of the imaging performance (see Supplementary Note 1 for the detailed experimental setup).

**Correction of position-dependent and high-order aberrations**. With the increase of the target depth, the aberration maps vary drastically from one area to another. In technical terms, an isoplanatic patch where an aberration correction map is effective becomes smaller as targets are located deeper within tissues. To guarantee a high resolving power over a wide view field, it is critical to compensate local position-dependent aberrations. Because the time-gated reflection matrix that we recorded has all the necessary information on the coherent interaction of light with the specimens of interest, we can perform image postprocessing to address local aberrations. To verify the performance of our method, we conducted brain imaging of a living zebrafish at three weeks after fertilization (see "Methods" section for the sample preparation details). Zebrafish have served as a useful model platform for studying the neural development of vertebrates, mainly owing to the rapid development and transparency of the embryos[33]. However, as a zebrafish ages, pigmented scales, complex skin architectures, and internal organs significantly deteriorate the resolving power of conventional high-resolution confocal imaging. For instance, dark area indicated by the yellow dashed box in Fig. 2d is the obscured image due to pigmented scales. Therefore, the investigation has mostly been focused on the early developmental stages up to 5–6 dpf for brain imaging[11,34], and the degradation of spatial resolution was inevitable for more matured zebrafish[34–36], and studies for later developmental stages require physical dissection. In our study, a 21-dpf larval zebrafish introducing both strong aberration and multiple light scattering was placed under the objective lens (Fig. 2d), and multiple time-gated reflection matrices were

recorded at various depths in the hindbrain. The center wavelength of the light source was set as 515 nm to achieve a high spatial resolution. Figure 2a shows an image before the aberration correction taken at a depth of 160 μm. This image was reconstructed from the diagonal elements of the time-gated reflection matrix, i.e., $E_S(\mathbf{r}^o = \mathbf{r}^i; \mathbf{r}^i, \tau_0)$. This is equivalent to the time-gated confocal microscope image, because the detection position $\mathbf{r}^o$ is the same as the illumination position $\mathbf{r}^i$. In this image, the bundles of neural fibers known as myelinated axons were observed, as their lipid-rich membranes have higher reflectivity than the surrounding area, but individual axons were hardly resolved due to the sample-induced aberrations. In fact, conventional confocal fluorescence/reflectance images were even worse than this image because of the lack of temporal gating (see Supplementary Note 8 for image comparison).

For the aberration correction, the full view field was divided into 6 × 6 segments, as indicated by the yellow dotted boxes in Fig. 2d. The sub-matrix for each segment was independently examined to correct the local sample-induced aberration. For instance, Fig. 2e shows $E_{sub}(\mathbf{r}^i)$ for the area indicated by a white arrow in Fig. 2a. The signals in this matrix were spread out from the diagonal to the off-diagonal elements, and the approximate point spread function (PSF) obtained from the intensity map of $E_{sub}(\mathbf{r}^o; \mathbf{r}^i, \tau_0)$ for a specific input position $\mathbf{r}^i$ was significantly broadened (Fig. 2h). We transformed the basis of this sub-matrix to the wavevector space, applied the aberration correction algorithm, and transformed the matrix back to the position space (see Methods for the detailed aberration correction procedure). Figures 2f and i show the aberration-corrected sub-matrix $E_{sub}^{cor}(\mathbf{r}^o; \mathbf{r}^i, \tau_0)$ and its PSF, where the signals that used to be spread out to off-diagonal elements were gathered together and concentrated on the diagonal elements. This differs from other software-based AO microscopy methods based on the confocal detection, in which the aberration correction does not gather the defocused energy back to the focus, as it is not detected in the first place. As shown in Fig. 2j, we plotted the line profiles of the PSF before and after the aberration correction and observed that the full width at half maximum of the PSF was reduced to 370 nm, which is the diffraction-limit PSF size for 0.8 NA. The enhancement of the Strehl ratio defined by the ratio of peak intensities after and before the aberration correction was 52 (see Supplementary Note 5 for detailed analysis). We repeated this operation of aberration correction for all the segments and obtained the aberration-corrected full-field image (Fig. 2b). The fine filament structures of individual myelinated axons are clearly resolved over the entire field of view. In some areas, such as the one indicated by a yellow arrow in Fig. 2b, even the fine neural fibers that used to be invisible appeared. Figures 2c and g show the aberration maps for each of the 6 × 6 segments and the normalized cross-correlation of these aberration maps with respect to the one marked by a black arrow in Fig. 2c, respectively. The aberration maps which differ significantly from one another indicates the spatial variability and complexity of the sample-induced aberrations.

**Aberration-free and high-resolution neuroimaging of living zebrafishes**. We conducted volumetric imaging for the hindbrain of two living larval zebrafishes, 6 and 10 dpf. A series of time-gated reflection matrices were recorded from dorsal to ventral to the depth of ~220 μm with a depth scanning step of 5 μm. After reconstructing an aberration-corrected slice image at each depth, numerical propagation was applied at an increment of 0.5 μm to fill the gap between the neighboring slices (see Supplementary movies 1 and 2 for the sequential depth-dependent images). Figure 3a shows the three-dimensional (3D) rendered image of a

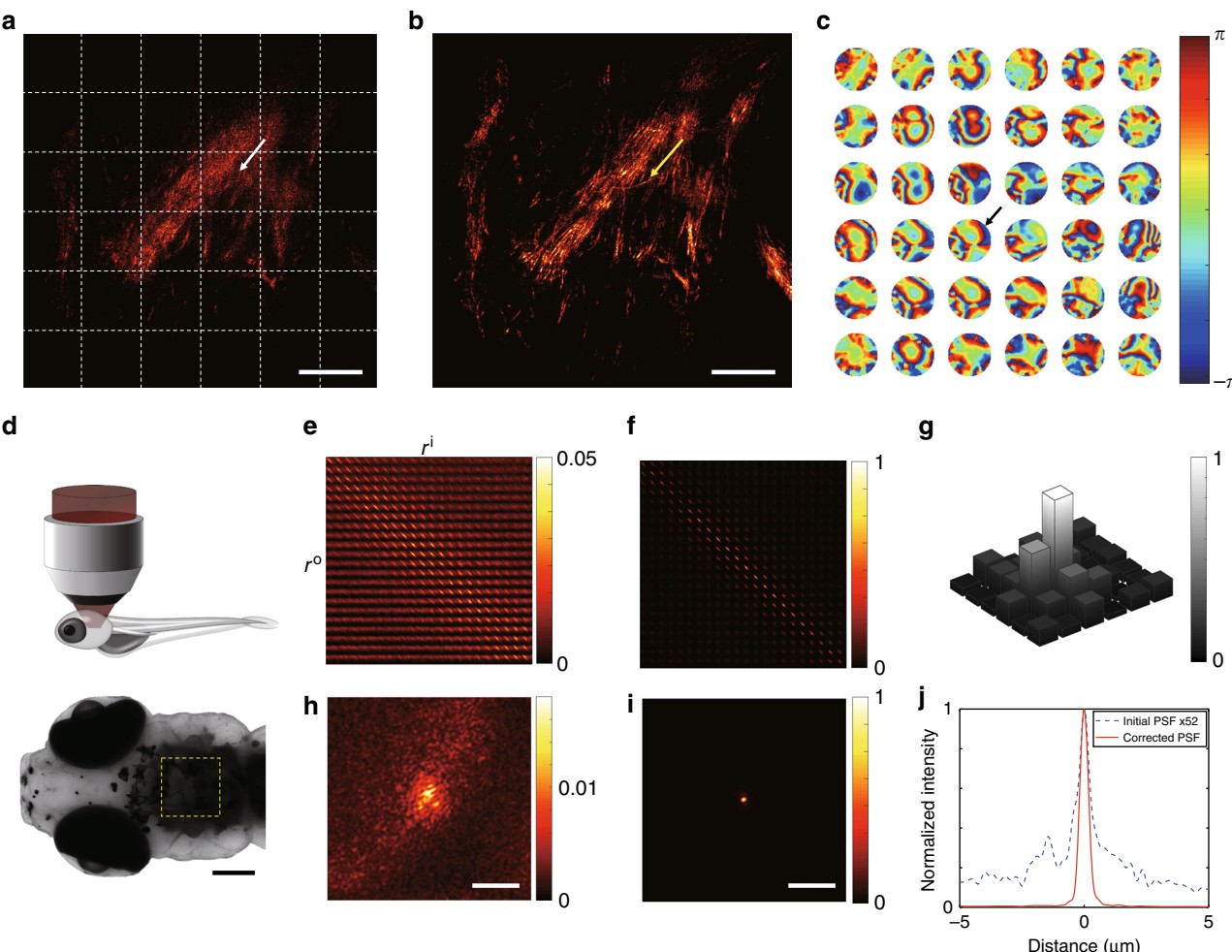

**Fig. 2** Correction of local position-dependent aberrations for in vivo neuroimaging. **a** Reconstructed image of SASM before the aberration correction. The white dotted boxes show segmented areas where aberration correction was individually applied. **b** Aberration-corrected image of AO-SASM. The scale bar represents 20 μm (**a**, **b**). **c** Local aberration maps in the pupil plane corresponding to the segments divided by the white dashed boxes in (**a**). Color bar, phase retardation in radians. The radius of each aberration map is $k_0 \times NA$, where $k_0$ is the magnitude of the free-space wavevector. **d** Imaging configuration. A 21-dpf zebrafish was placed under the objective lens at an upright position after being anesthetized, and the area close to the ear in the hindbrain was investigated. The dorsal view of the zebrafish taken by a bright-field microscope is shown below. The dark area in the yellow dashed box is due to the pigmented scales at the skin. The scale bar represents 200 μm. **e**, **f** Amplitude maps of the time-gated reflection matrices for the segment indicated by a white arrow in (**a**) before and after aberration correction, respectively. **g** Normalized cross-correlation between aberration maps in their complex pupil functions with respect to the aberration map indicated by a black arrow in (**c**). **h**, **i** PSFs derived from (**e**, **f**), respectively. The color bars in (**e**, **f**, **h**, **i**) represent the intensity normalized by the maximum value of the corrected PSF in (**i**). The scale bar represents 5 μm (**h**, **i**). **j** Line profiles of PSFs obtained from **h** (blue dotted curve) and **i** (red solid curve). The PSF from (**h**) was multiplied by the factor of 52

10-dpf zebrafish, which was acquired from the reconstructed slice images for the imaging volume of $220 \times 220 \times 150$ μm composed of 43,200 isoplanatic patches of $18 \times 18 \times 0.5$ μm. To investigate the detailed neuroanatomy, the depth range of the volumetric image was decomposed into a few different parts and the maximum intensity projection (MIP) image for each part was displayed in Fig. 3b–e. At the upper part of the volume, the en-face cellular distribution (Fig. 3b) is clearly shown between lobus caudalis cerebella (LCa) and crista cerebellaris (CC) in the depth range of 70–75 μm. Underneath CC, the neural network encompassing reticulospinal neuron projections are shown including those of the crossed Mauthner neurons and the series of seven ladder-like commissural tracts in the caudal hindbrain at the depth between 112.5 and 135 μm (Fig. 3c). Figure 3d, e shows the dorsal and ventral network comprising medial and lateral (blue arrows) projections spanning posterior and anterior sites. They revealed two prominent commissural fibers in rhombomere

3 (yellow arrows), which are known to play an important role in controlling fast turning movements in conjunction with Mauthner neurons. In particular, we observed that the anterior medial projections are extended to the two remarkably symmetric and tightly fasciculate bundles with Y shape, and the left- and right-hand parts of the medial projections are reciprocally connected to fine commissural tracts indicated by white arrowheads (Fig. 3e). Furthermore, the interneuron projections (magenta arrowheads) were clearly resolved, which transfer information to central neurons from the torus semi-circularis (magenta arrows) functionally related to auditory stimuli in the inner ear (see individual slice images in the Supplementary movies 1 and 2 for detailed identification of the neuroanatomy). Notably, these precise visualizations of structural details are unprecedented in the context of label-free in vivo imaging of a zebrafish.

As a control experiment, we took confocal fluorescence images that have widely been used in previous studies[35,36] and compared

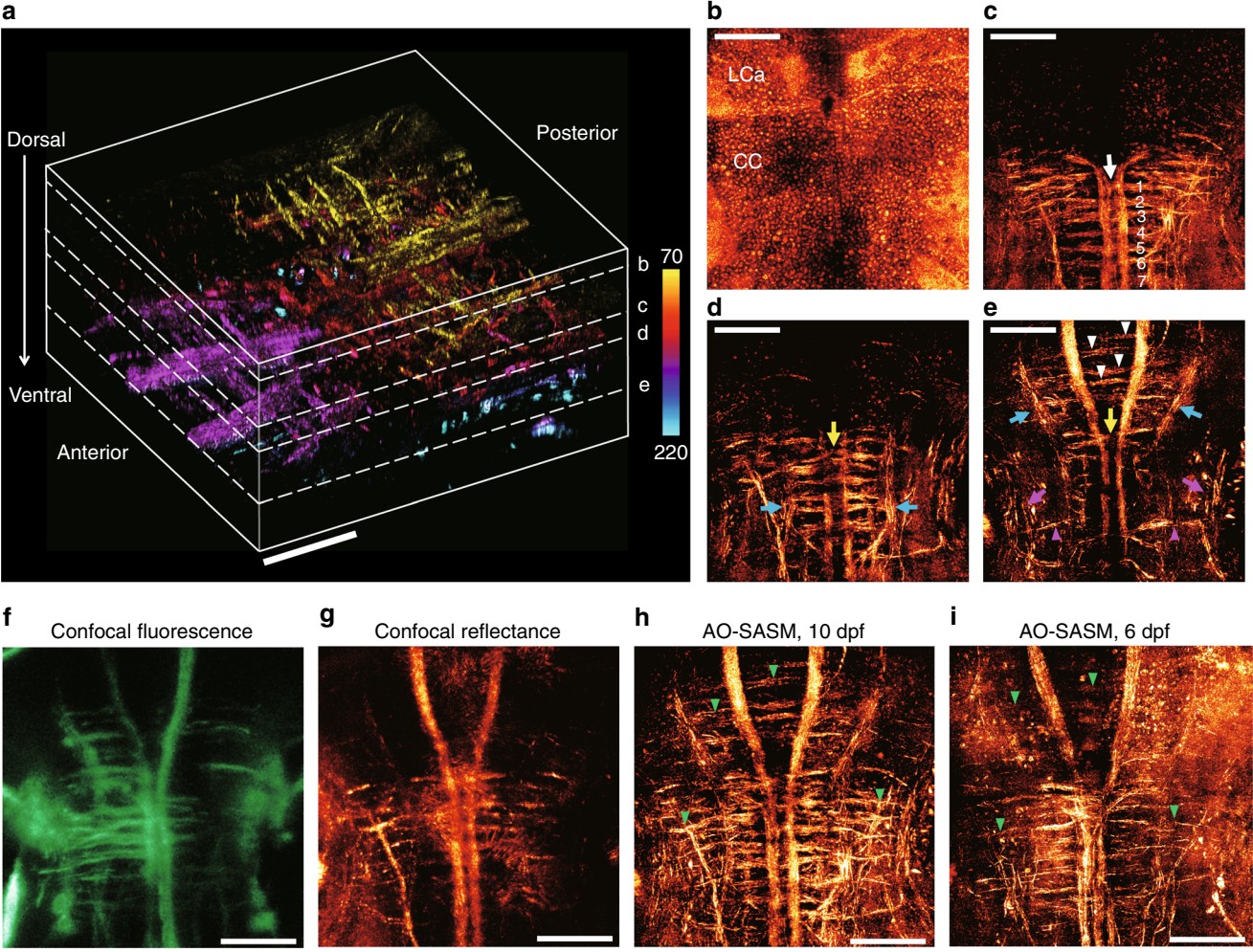

**Fig. 3** 3D neuroanatomy encompassing the deep hindbrain of a living zebrafish. **a** 3D rendering of the aberration-corrected tomographic images ranging from 70 to 220 μm in depth at the hindbrain of a 10-dpf larval zebrafish. The wavelength of the light source was 633 nm. The scale bar represents 50 μm. Color bar indicates depth from the surface. **b**–**e** MIP images at a depth centered on each plane indicated by the white dashed lines in (**a**). The scale bar represents 50 μm. **b** The en-face cellular distributions. MIP image ranging from 70 to 75 μm in dorsal depth. **c** The subset of neural networks. MIP image in the depth range of 112.5 to 135 μm. The white arrow indicates the crossed axons of Mauthner cells, and the labels ranging from 1 to 7 indicate a series of seven ladder-like commissural tracts in the caudal hindbrain. **d** MIP image ranging from 135 to 145 μm. **e** MIP image ranging from 150 to 220 μm. The yellow arrows in (**d**, **e**) indicate two prominent commeasures in rhombomere 3. The blue arrows indicate the ipsi- and contra-lateral projections. The white arrowheads in (**e**) indicate commissural tracts connected to the anterior medial projections. The magenta arrows and arrowheads in **e** indicate torus semi-circularis and interneuron projections, respectively. **f**–**h** MIP images of confocal fluorescence (**f**), confocal reflectance (**g**), and AO-SASM (**h**) for the same zebrafish shown in (**a**). **i** MIP image retrieved by AO-SASM for a 6-dpf larval zebrafish. Green arrowheads in (**h**, **i**) indicate commissural tracts connecting between medial and lateral projections. The scale bar represents 50 μm. LCa lobus caudalis cerebella, CC crista cerebellaris

them with AO-SASM images. Volumetric confocal fluorescence images and their MIP image at dorsal view (Fig. 3f) were acquired for the same area of the zebrafish shown in Fig. 3a. Because the zebrafishes used in our study were Tg(*claudinK:gal4;uas:mgfp*) larvae in which green fluorescent proteins are expressed at the membrane of oligodendrocytes that wrap around the axons, fluorescence signals effectively visualize the myelin that surrounds axons. For comparison, the MIP image of the AO-SASM image over the entire depth range of Fig. 3a is shown in Fig. 3h. The similarity of the overall structures to those identified from AO-SASM confirms that the myelin sheath is the main source of reflectance signals. However, the fine myelin processes were unresolved in the confocal fluorescence images owing to the pronounced sample-induced aberration. Confocal reflectance microscopy was also used in recent years to investigate myelin process owing to its label-free imaging capability[37–39]. Similar to

the confocal fluorescence imaging, only dense fascicles were visible, and individual axonal branches remained missing in the volumetric confocal reflectance images taken at the same area (Fig. 3g). In addition, we took an AO-SASM image of a 6-dpf zebrafish (Fig. 3i) and compared it with that of a 10-dpf zebrafish (Fig. 3h) to observe the longitudinal development of the oligodendrocyte process. While their basic scaffold is homologous between the different developmental stages, increased populations and fasciculation of neural fibers proved the significant progress of the neural development. Especially, the commissural tracts marked by green arrowheads connecting between the two medial and lateral projections exhibited distinct development in the 10-dpf zebrafish than the 6-dpf zebrafish. All these results indicate that AO-SASM can map the detailed neuroanatomy to a greater depth and for a more matured developmental stage than the conventional imaging modalities.

## Discussion

We presented a high-speed and label-free adaptive optical coherence imaging method that is fast enough to perform in vivo imaging. The internal structures of a living zebrafish were visualized with a diffraction-limit spatial resolution (370–480 nm) up to the depth where conventional imaging modalities fail. By dealing with both position-dependent sample-induced aberrations and multiple-scattering noise with no need for guiding stars and hardware feedback, AO-SASM noninvasively offered the atlas of the nervous system and the tomography of fine myelinated axon projections inside the hindbrain of a larval zebrafish in its native state. Recent studies demonstrated that the quantification of demyelination and remyelination processes is critical for assessing the physiological states of the nervous systems[38,40]. While confocal fluorescence microscopy was most widely used in the past[34–36], label-free imaging methods such as confocal reflectance microscopy[37–39] and optical coherence microscopy[41,42] have drawn attention in recent years for their accuracy in the quantification of myelinated axons and reliability in longitudinal studies. While these investigations were mostly focused on the early developmental stages up to 5–6 dpf in the case of a zebrafish owing to their imaging depth limit, the proposed method allowed us to visualize the whole neural network of CNS in the hindbrain of the zebrafish even as matured as 21 dpf.

In recent years, the application of multi-photon microscopy such as two/three-photon fluorescence microscopy and second/third-harmonic-generation microscopy for studying a zebrafish has drawn much attention due to its deep penetration through scattering tissues and diverse contrast mechanisms[43,44]. While our AO-SASM visualizes myelinated axons and cellular compartments having higher reflectivity than surrounding tissues with the use of elastic backscattering, fluorescence microscopy can visualize specific types of cells or molecules at the expense of special sample preparations. And often times, its volumetric imaging rate could be dramatically improved in the form of light-sheet microscopy[45] and light-field microscopy[46]. And second- and third-harmonic-generation microscopy, while high excitation power is required, visualizes morphological context such as collagen fibers, skin cells, blood cells and lipid accumulations without the use of labeling agents. While these multi-photon microscopy techniques provide valuable molecular contrast, they suffer from the loss of spatial resolution due to the sample-induced aberrations especially when imaging zebrafishes. Unlike mouse brain tissues, many heterogeneous structures such as skin and internal organs in the zebrafish act as sources of aberration. According to the literatures reporting the use of multi-photon microscopy for studying the central nervous system of the zebrafish, the imaging depth was much shallower than 200 μm, and studies were limited to early developmental stages. On the contrary, AO-SASM reached the depth of 220 μm through a thick hindbrain with the spatial resolution close to the ideal diffraction limit. Therefore, the imaging depth of our method is better than or comparable to the state-of-the-art multi-photon microscopy if spatial resolution is accounted for in estimating the imaging depth. In this respect, AO-SASM can not only complement these existing imaging modalities in terms of information content, but also assist them to cope with the sample-induced aberration.

Irradiance of illumination used in AO-SASM was below 100 μW/mm², which is an order of magnitude lower than maximum permissible exposure level of 200 mW/cm² in animal tissues. It requires relative weak excitation beam in comparison with multi-photon microscopy as elastic backscattering is used for imaging. Furthermore, wide-field detection configuration is used instead of pinhole gating, which makes the signal collection more efficient. The image acquisition speed per depth, mainly determined by the camera frame rate, was improved to 4 Hz in the present implementation and can further be increased to a video-rate if a commercially available high-speed complementary metal–oxide semiconductor camera is used. This will extend the proposed method to in vivo imaging of animal and human subjects for various applications, including myelin-associated physiology in neuroscience, retinal pathology in ophthalmology, and endoscopic interrogation of internal organs. With all these benefits, our method expands the scope of studies and diagnosis that optical imaging can offer.

## Methods

**Detailed working principle of AO-SASM.** In the experimental setup shown in Fig. 1a, a two-axis GM was installed at the conjugate plane to the sample plane for the scanning of $\mathbf{k}^i$. It was placed before a beam splitter (BS1) dividing the laser beam into sample and reference arms for the synchronous scanning of sample and reference waves. For the off-axis interference imaging, a diffraction grating (DG, Edmund Optics, 120 lp/mm) was placed in the reference beam path, and its first-order diffraction was used as a reference wave. Therefore, the transverse wave-vector of the reference wave, $\mathbf{k}^i_R$, is given by $\mathbf{k}^i + \mathbf{k}_{DG}$, where $\mathbf{k}_{DG}$ is set by the pitch and orientation of the diffraction grating. The diffraction grating does not affect the temporal pulse front due to its uniform thickness. We sequentially scanned the angle of the GM to control the $\mathbf{k}^i$ of the plane wave incident to the sample and recorded interferograms of the backscattered waves. By taking the Hilbert transform of the recorded interferogram with respect to $\mathbf{k}_{DG}$, we obtained the complex field maps of the backscattered waves. The target depth was selected by adjusting the temporal gating window using a reference mirror (RM). Confocal reflectance and fluorescence microscopy were constructed on the top of the AO-SASM for the comparison of the imaging performance (see Supplementary Note 1 for the detailed experimental setup).

**Construction of a time-gated reflection matrix.** The main difference of AO-SASM from the conventional interference microscope and our earlier implementations[31,32,47] is that the reference wave rotates along with the sample wave. We must account for this difference before applying the aberration correction algorithm, which requires complex field maps in the laboratory frame. Let us consider a backscattered wave $E_S(\mathbf{r}^o; \mathbf{r}^i, \tau_0)$ from a sample for an incident wave, $E_{in}(\mathbf{r}, z = 0; \mathbf{k}^i) = E_S^0 \exp[-i\mathbf{k}^i \cdot \mathbf{r}]$, with a wavevector $\mathbf{k}^i$. Here, $\mathbf{r}_o$ is a spatial coordinate at the sample plane conjugate to the camera, and $\tau_0$ the gated time set by the reference mirror. We introduced a reference wave, $E_R(\mathbf{r}_o) = E_R^0 \exp[-i(\mathbf{k}^i + \mathbf{k}_{DG}) \cdot \mathbf{r}_o]$, with a wavevector of $\mathbf{k}^i + \mathbf{k}_{DG}$ to the camera to record an interferogram given by $I(\mathbf{r}_o; \mathbf{k}^i, \tau_0) = |E_S(\mathbf{r}_o; \mathbf{k}^i, \tau_0) + E_R(\vec{r}_o)|^2$. Under the condition that $|\mathbf{k}_{DG}|$ is larger than spatial frequency bandwidth of $E_S(\mathbf{r}_o; \mathbf{k}^i, \tau_0)$ set by the NA of the objective lens, we obtain the interference term

$$E_{GM}(\mathbf{r}_o; \mathbf{k}^i, \tau_0) = E_S(\mathbf{r}_o; \mathbf{k}^i, \tau_0)\left(E_R^0 \exp[-i\mathbf{k}^i \cdot \mathbf{r}_o]\right)^*$$

in $I(\mathbf{r}_o; \mathbf{k}^i, \tau_0)$ via the Hilbert transform of $I$ with respect to $\mathbf{k}_{DG}$. $E_{GM}(\mathbf{r}_o; \mathbf{k}^i, \tau_0)$ is the complex field map of the backscattered wave in the frame of the rotating reference wave, as it contains the term, $\left(E_R^0 \exp[-i\mathbf{k}^i \cdot \mathbf{r}_o]\right)^*$. For the reconstruction of an object image and application of the aberration correction algorithm, we must extract $E_S(\mathbf{r}_o; \mathbf{k}^i, \tau_0)$, which is the complex field map in the laboratory frame. We separately measured $\mathbf{k}^i$ by using a diffusive scattering sample and normalized out $\left(E_R^0 \exp[-i\mathbf{k}^i \cdot \mathbf{r}_o]\right)^*$ in $E_{GM}(\mathbf{r}_o; \mathbf{k}^i, \tau_0)$ for each $\mathbf{k}^i$ (see Supplementary Note 3 for the conversion to the laboratory frame). Using the basis transform of $E_S(\mathbf{r}_o; \mathbf{k}^i, \tau_0)$ with respect to $\mathbf{k}^i$, we obtained a time-gated reflection matrix in the position basis (see Supplementary Note 4 for construction of the time-gated reflection matrices) whose sub-matrix $E_{sub}(\mathbf{r}_o; \mathbf{r}_i, \tau_0)$ is shown in Fig. 2e. Here $\mathbf{r}_i$ is conjugate to $\mathbf{k}^i$ such that it corresponds to the position of illumination. From this matrix, we can extract the PSF from the intensity map across $\mathbf{r}_o$ at a selected value of $\mathbf{r}_i$ (see Supplementary Note 5 for the PSFs before and after the application of the aberration correction algorithm).

**Application of aberration correction algorithm to the time-gated reflection matrix.** We employed AO correction to a sub-matrix $E_{sub}(\mathbf{r}_o; \mathbf{r}_i, \tau_0)$ for correcting local sample-induced aberrations. Because aberration correction needs to be applied on the pupil plane, we converted $E_{sub}(\mathbf{r}_o; \mathbf{r}_i, \tau_0)$ to $E_{sub}(\mathbf{k}^o; \mathbf{k}^i, \tau_0)$ via the basis transformation. We then applied the aberration correction map $\theta_i^{(1)}(\mathbf{k}^i)$ to each $\mathbf{k}^i$ column of the matrix in such a way to maximize the total intensity, i.e. the sum of pixel intensities in the image, of $E_{sub}(\mathbf{r}_o = \mathbf{r}_i; \mathbf{r}_i, \tau_0)$. With $\theta_i^{(1)}(\mathbf{k}^i)$ in place, we then applied the aberration correction map $\theta_o^{(1)}(\mathbf{k}^o)$ to each $\mathbf{k}^o$ row of the corrected $E_{sub}(\mathbf{k}^o; \mathbf{k}^i, \tau_0)$ for maximizing the total image intensity. The applications of the correction maps $\theta_i^{(1)}(\mathbf{k}^i)$ to each $\mathbf{k}^i$ and $\theta_o^{(1)}(\mathbf{k}^o)$ to each $\mathbf{k}^o$ form a separate

treatment of input and output aberrations. The steps were iterated $n$ times to find $\theta_i^{(n)}(\mathbf{k}^i)$ and $\theta_o^{(n)}(\mathbf{k}^o)$ until the total image intensity of $E_{sub}(\mathbf{r}_o = \mathbf{r}_i; \mathbf{r}_i, \tau_0)$ was converged. Typically, 5–10 iterations were sufficient for the convergence, and the computation time was approximately 10 s. The sample-induced aberration was acquired by adding correction maps of the entire iterations, i.e., $\theta(\mathbf{k}^i) = \sum_{j=1}^{n} \theta_i^{(j)}(\mathbf{k}^i)$ or $\theta(\mathbf{k}^o) = \sum_{j=1}^{n} \theta_o^{(j)}(\mathbf{k}^o)$ (Fig. 2c). After the application of the aberration correction algorithm, the signals were concentrated on the diagonal elements, and the width of the PSF was sharpened to be 370 nm, which was the diffraction-limit spatial resolution of the system. See Supplementary Note 5 for the detailed steps of the aberration corrections.

**Preparation of zebrafish samples**. Zebrafish Tg(*claudinK:gal4;uas:mgfp*) embryos were raised at 28 °C in an E3 embryo medium (5 mM NaCl, 0.17 mM KCl, 0.33 mM CaCl₂, 0.33 mM MgSO₄). First, 24-h post fertilization, they were transferred to the E3 medium containing N-phenylthiourea (Sigma) for inhibiting pigmentation. After 6 to 21 dpf, zebrafishes were anesthetized by adding tricaine (Sigma) to the E3 medium. They were then immersed in 1.5% low-melting agarose (invitrogen) and mounted on a slide glass. We supplied the mounted specimen with the E3 medium containing tricaine to maintain anesthesia during the investigation under the microscope. The immersion solution was continuously warmed by a temperature controller (TC200; Thorlabs, USA) to maintain the temperature at 24–26 °C. All animal experiments were approved by the Korea University Institutional Animal Care & Use Committee.

## Data Availability
The data supporting this study are available from the corresponding author upon reasonable request.

## Code Availability
The code to analyze the data is available from the corresponding author upon reasonable request.

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

## Acknowledgements
This research was supported by IBS-R023-D1, the National Research Foundation of Korea (2019R1C1C1008175, 2016R1A6A3A11936389, 2016R1D1A1B03933770), the Catholic Medical Center Research Foundation made in the program year of 2018, and the KERI Primary research program of MSIT/NST (No. 18-12-N0101-41). We are grateful to Myunghwan Choi for constructive discussion.

## Author contributions
M.K., Y.J, S.Y., S.Kang, and W.C. conceived and designed the experiment. M.K. and Y.J. constructed the experimental setup and performed measurements with J.H.H. The experimental data were analyzed by M.K., Y.J., J.H.H. and W.C. S.Kim H.-C.P. J.H.H. prepared the zebrafish samples and provided discussion on image interpretations. K-D.S. fabricated resolution targets. B.L. and G.H.K. supported the lasers and optical design. M.K., Y.J., J.H.H. and W.C. prepared the manuscript, and all authors contributed to finalizing the manuscript.

## Additional information

**Competing interests:** The authors declare no competing interests.

