## [Peer Review File · Nature Communications]

Reviewers' comments:

Reviewer #1 (Remarks to the Author):

Kim et al report on the use of a high-speed adaptive optics microscope that uses intrinsic elastic backscattering as a source of contrast for deep-tissue imaging, illustrated with images of living zebrafish. The method is very elegant and the images that are obtained are impressive, and would in principle merit publication in Nature Communications. I am not 100% up to date with the forefront of interferometric imaging, so will not comment on the combined novelties in obtaining the images, except that the results obtained are very good and at high image acquisition speed. The authors fail however to put their results in the correct context. For instance, in the last sentence of the abstract and introduction they refer to 'conventional imaging modalities'. Only in the discussion it becomes clear which conventional methods they refer to: confocal fluorescence microscopy, reflectance and optical coherence microscopy, which "mostly focused on the early developmental stages up to 5–6 dpf in the case of a zebrafish owing to their imaging depth limit". The authors should refer here also to nonlinear labelled and label free microscopy, which can easily reach the depth obtained here (of 220 microm). For instance, the group of Beaurepaire, who works mainly on zebrafish too, image up to a depth of 500 microm in 3-months (instead of 21 dpf) old zebrafish (Guesmi et al. Light: Science & Applications (2018) 7:12) using 3th harmonic generation microscopy. One can then compare and discuss the information content of the images, but this is the standard they have to relate to.

Minor points:

Why do the authors refer to 'origins of disorder' as motivation for label free in vivo imaging?

What is the time associated with the image processing steps?

Reviewer #2 (Remarks to the Author):

High resolution, in vivo, deep tissue imaging is very important but also challenging. This paper proposed a new experimental configuration to achieve higher image acquisition rate compared to their previous SLM-based method. Here are my comments:

1. The authors need to explain more details about how to distinguish the multiple scattering and aberration. What are the differences between the processing? Did the authors extract the single scattering signal first and then apply the aberration correction? In the case when aberrations present, the PSF of a single scatterer would also spread out on the detector plane. Also, how to suppress the interruption of speckles?

2. From lines 69-71, the authors claimed that most of the developed software-based AO algorithms required guiding stars, which is not correct. Examples can be found from Opt. Express 21(9), 10850 (2013), Opt. Lett. 41(14), 3324–3327 (2016), Sci. Rep. 6, 35209 (2016)

3. From lines 134-136, "The typical bandwidth used in the experiment was 15 nm. This corresponds to the temporal coherence time of approximately 100 fs, which is the effective width of the pulse front." First of all, the direct output from the supercontinuum generation is not possible at the femtosecond level (the author didn't define "fs" as well) because of the chirping. The authors should use the coherence/time gate here instead. The axial resolution should be provided. Secondly, the output bandwidth of the light source could be much larger than 15 nm? What was the reason to choose only 15 nm? Larger bandwidth provides better axial resolution. Is there any experimental limitation for using a larger bandwidth?

4. When scanning the different angles of illumination, for each recording, the back-scattering light came from different volumes and different angles related to the camera plane. How to make sure the synthetic information from the same sample plane along depth when imaging a continuous

tissue sample? As the reference beam was also scanning, there was no constant reference plane (not the referenced beam) at a specific depth.

5. How is the performance of rejecting multiple scattering by comparing this matrix method to a physical pinhole (e.g. fiber) in point-scanning OCT? Although the authors claimed that “where the signals that used to be spread out to off-diagonal elements were gathered together and concentrated on the diagonal elements. This differs from other software based AO microscopy methods based on the confocal detection, in which the aberration correction does not gather the defocused energy back to the focus...” This sentence was only discussing the aberrations. The example of confocal imaging was not enough as it didn't provide the coherence gate. I suspect the dynamic range would be decreased compared to common point-scanning FD-OCT as all the useless light from other planes would enter the camera. What is the SNR for the system?

6. As off-axis geometry was used, the authors should also explain that band-pass filter in the Fourier-domain was used before all the other processing.

7. As there was no physical pinhole, how deep this technique can image? How about comparing it to a common FD-OCT/OCM with hardware-/software-based AO?

8. What was the volumetric rate? It seems this technique is still quite slow, and the in vivo sample has to be fixed.

9. Because there was no actual wavefront modulator used, the title of AO might cause some confusion. I would suggest changing it to computational or digital AO.

Reviewer #3 (Remarks to the Author):

This paper presents a label-free method for in vivo neural imaging in the larval zebrafish brain. The method is based on a synchronous angle scanning approach, known by the acronym 'CASS' that has previously been presented by the authors (Ref. 28). The compensation of aberrations in combination with CASS has also been demonstrated (Ref. 29). Thus the main novelty of this paper is the application these methods to in vivo neural imaging in the larval zebrafish brain.

Since the authors have previously published on the technical aspects of CASS with aberration correction, the significance of the current work should be evaluated with respect to the state-of-the-art imaging approaches for in vivo imaging in the zebrafish and other animal brain models. Relevant papers based on lightsheet, multiphoton microscopy, and other novel confocal approaches such as SCAPE should be cited, and the advantages/disadvantages of the current method should be discussed in the paper. In particular, how do the capabilities of the method compare to the volumetric imaging speed, imaging depth, and overall information content of previous methods, including the OCT work that the authors have already referenced in the paper? For example, Prevedel et al., *Nature Methods*, 11:7, 2014 and Bouchard et al, *Nature Photonics*, Vol 9, 2015 have demonstrated volumetric imaging with 20 Hz update rate and faster. Three-photon microscopy has demonstrated deep imaging in mouse brain with cellular resolution (Ouzounov et al, *Nature Methods*, Vol. 14, 2017). Compared to these previous approaches for neuroscience imaging studies, what advantages/disadvantages does the present label-free approach offer?

A key concern for in vivo CASS imaging is motion of samples, e.g. from heart beat, respiration, etc. I suspect that what the authors refer to as 'sample-induced aberrations' is not just due to sample-induced aberrations in the traditional sense, but also could contain a significant contribution from the in vivo motion of samples. Sample motion will disrupt the phase of the coherently accumulated signal acquired from multiple illumination angles, since the plane wave

components contributing to the PSF, such as shown in Fig 2j, are acquired in a multiplexed fashion as a function of time, rather than simultaneously for imaging at the focal plane of a focused beam. I suspect that the larval zebrafish is quite a stable sample, but can the authors provide some numbers to quantify the level of phase instability introduced by this type of sample, and add a discussion on impact of sample motion for in vivo applications?

Lines 174-175 state that a 21-dpf larval zebrafish introducing both strong aberration and multiple scattering was used in this study. What evidence can the authors provide to support their claim about multiple scattering? Are they able to estimate the imaging in terms of the number of scattering lengths for the results in Fig. 2?

The authors state that they have achieved a record effective aberration correction speed of 10,000 modes/s. Can the authors include a justification for this number in the main paper or supplementary information? Since the authors are doing a software aberration correction in post processing, why is the "effective aberration correction speed" that they calculated a fair comparison to traditional AO methods? What was the total computation time required to reconstruct aberration corrected en face images and for the whole reconstructed volume? Also, can the authors provide some evidence that a correction with 10k modes were required (otherwise it is meaningless to try and perform this with traditional AO methods), and therefore why this is a fair comparison to hardware correction methods?

The authors state that the number of angles N_{in} can be significantly less than the number of modes in the field of view. However, I think that acquiring fewer measurements should have an impact on the information content of the reconstructed image. A fewer number of angles would likely reduce the field of view of the reconstructed image due to a drop of signal at the edges of the field of view. Can the authors add a discussion on the trade-offs involved with acquiring lower number of angles?

What was the overall volumetric imaging time or rate? Only the en face update rate of 4Hz was reported. Was the depth sampling equal to the coherence length of the source, and so how many planes were acquired along the depth axis?

Strehl ratio is normally given as a number between 0 and 1, whereas Lines 203-204 a Strehl ratio of 52 is reported. If the authors cannot give the actual Strehl ratio, which compares to the ideal diffraction limited spot, then it may be better to report the factor by which the Strehl ratio improved.

In the abstract the authors say that they were able to visualize "anatomical details including fine neuronal processes". I would suggest that the authors do not use the phrase "neuronal processes" to describe anatomical structure, since "processes" give the impression of time-lapse monitoring of a biological function.

In the Supplementary Info: It will be helpful to provide additional details of the optical system in fig. S1, such as focal length/part numbers of lenses, telescope magnifications, etc. Also, line 115 states that the AO algorithm maximizes "total intensity of the reconstructed image". Doesn't an aberrated PSF have the same energy as a computationally refocused PSF, just that the energy is blurred out in space? Can the authors clarify if they use sum of pixel intensities in the image or mean peak reconstructed intensity?

Reviewers' comments:

Reviewer #1

Kim et al report on the use of a high-speed adaptive optics microscope that uses intrinsic elastic backscattering as a source of contrast for deep-tissue imaging, illustrated with images of living zebrafish. The method is very elegant and the images that are obtained are impressive, and would in principle merit publication in Nature Communications. I am not 100% up to date with the forefront of interferometric imaging, so will not comment on the combined novelties in obtaining the images, except that the results obtained are very good and at high image acquisition speed.

We appreciate reviewer's thinking highly of our imaging results. Indeed, our method presents high-resolution label-free imaging working in the significantly aberrating and scattering specimens. As we explained in detail in the following, our synchronous angular scanning interferometry is a new technical development in the context of interferometric microscopy, and it played a critical role in recording a unique time-gated reflection matrix with a greatly improved imaging speed relevant to *in vivo* imaging.

The authors fail however to put their results in the correct context. For instance, in the last sentence of the abstract and introduction they refer to 'conventional imaging modalities'. Only in the discussion it becomes clear which conventional methods they refer to: confocal fluorescence microscopy, reflectance and optical coherence microscopy, which "mostly focused on the early developmental stages up to 5–6 dpf in the case of a zebrafish owing to their imaging depth limit". The authors should refer here also to nonlinear labelled and label free microscopy, which can easily reach the depth obtained here (of 220 microm). For instance, the group of Beaurepaire, who works mainly on zebrafish too, image up to a depth of 500 micron in 3-months (instead of 21 dpf) old zebrafish (Guesmi et al. Light: Science & Applications (2018) 7:12) using 3th harmonic generation microscopy. One can then compare and discuss the information content of the images, but this is the standard they have to relate to.

We considered confocal fluorescence/reflectance microscopy and optical coherence microscopy as conventional imaging modalities for the following reasons. Since our method uses elastic backscattering as a source of contrast, we compared its performance with confocal reflectance microscopy and optical coherence microscopy, both relying on the same contrast mechanism. In addition, we included confocal fluorescence microscopy as a point of reference since it is the most widely used modality in studying zebrafish. The reviewer suggested that multi-photon microscopy should be included in the conventional imaging modalities. We agree with the reviewer's opinion and thus revised the text accordingly. And we defined the conventional imaging modalities at the early part of manuscript by adding the following sentence to the introduction section.

"In the conventional imaging modalities such as confocal fluorescence/reflectance microscopy, optical coherence microscopy and multi-photon microscopy, these structures have been inaccessible in the matured stage without the dissection of the specimens due to the severe sample-induced aberrations."

The contrast mechanism of multi-photon microscopy such as two-photon/three-photon fluorescence microscopy and second/third-harmonic-generation microscopy is different from our method. Therefore, these imaging modalities carry different information from the proposed method. The following paragraph comparing AO-SASM with multi-photon microscopy is

added to the discussion section.

“In recent years, the application of multi-photon microscopy such as two/three-photon fluorescence microscopy and second/third-harmonic-generation microscopy for studying a zebrafish has drawn much attention due to its deep penetration through scattering tissues and diverse contrast mechanisms (Ouzounov et al, Nature Methods, Vol. 14, 2017, Light: Science & Applications (2018) 7:12). Fluorescence microscopy can visualize specific types of cells or molecules at the expense of special sample preparations, and often times its volumetric imaging rate could be dramatically improved in the form of light-sheet microscopy (Bouchard et al, Nature Photonics, Vol 9, 2015) and light-field microscopy (Prevedel et al., Nature Methods, 11:7, 2014). And second- and third-harmonic-generation microscopy, while high excitation power is required, visualizes morphological context such as collagen fibers, skin cells, blood cells and lipid accumulations without the use of labeling agents. On the other hand, our AO-SASM visualizes myelinated axons and cellular compartments having higher reflectivity than surrounding tissues with the use of elastic backscattering. AO-SASM can not only complement these existing imaging modalities in terms of information content, but also assist them to cope with the sample-induced aberration. The degradation of the spatial resolution of multi-photon microscopy is inevitable due to the strong sample-induced aberrations in the case of zebrafish, and AO-SASM can potentially provide a solution to correct such aberrations.”

The reviewer mentioned that nonlinear labelled and label-free microscopy can easily reach a depth deeper than 200 μm by providing the previous work by Guesmi et al. (Light: Science & Applications (2018) 7:12) as an example. This study performed imaging with three-photon and third-harmonic-generation microscopes for a three-month-old adult zebrafish, but the imaging area was different from our study. Images were acquired through a relatively thin dorsal view free from pigmentations and scales. It showed neural cells in telencephalon, not myelinated axons in the central nervous system. Therefore, direct comparison of the ages of zebrafish is not relevant. Furthermore, the maximum imaging depth was 237 μm , not 500 μm , for the zebrafish imaging. The imaging depth of 500 μm was obtained for a relatively homogeneous mouse brain tissue. At this depth of 237 μm , a single-cell level of spatial resolution was obtained, not the diffraction limit resolution. We conjecture that sample-induced aberration played a role in the degradation of the spatial resolution. On the contrary, AO-SASM reached the depth of 220 μm through a thick hindbrain with the spatial resolution close to the ideal diffraction limit. Therefore, the imaging depth of our method is better than or comparable to the state-of-the-art multi-photon microscopy if spatial resolution is accounted for in estimating the imaging depth.

We would like to stress that it is not straightforward to reach the imaging depth of 200 μm with diffraction-limit spatial resolution in the case of zebrafish due to the severe sample-induced aberration. Unlike mouse brain tissues, many heterogeneous structures such as skin and internal organs in the zebrafish act as sources of aberration. In fact, when we surveyed vast amount of literatures reporting the use of multi-photon microscopy for studying the central nervous system of the zebrafish, the imaging depth was much shallower than 200 μm , and studies were limited to early developmental stages.

Minor points:

Why do the authors refer to ‘origins of disorder’ as motivation for label free in vivo imaging?

The origins of disorder of biological systems such as neurological disorder can be investigated by either the labelled or label-free imaging. However, the important motivation of the label-free *in vivo* imaging is to investigate biological disorder in *most native states*. Although recent labeling techniques are so matured as to be minimally invasive, the value of label-free imaging still holds. This is especially the case in biomedical applications.

What is the time associated with the image processing steps?

The computation time depends on the field of view and the number of segments. When a personal computer with CPU clock speed of 3.6 GHz was used, it took about 20~30 minutes for reconstructing the coherent volume of $110 \times 110 \times 15 \mu\text{m}^3$. The iterative aberration correction for a single segment of $22 \times 22 \mu\text{m}^2$ took about an order of ten seconds. It took about tens of minutes to calculate *en face* image of the full field of $110 \times 110 \mu\text{m}^2$, which is composed of 6×6 segments. In addition, it took about ten minutes to computationally propagate the acquired *en face* image to 10 different depths within the coherent volume of $110 \times 110 \times 15 \mu\text{m}^3$. Total computation time for a whole central nerve system encompassing the hindbrain of the zebrafish shown in Fig. 3, which consists of 120 coherent volumes or 43,200 segments, was around 50 hours. However, there is much room to optimize the computation since the current algorithm is sequentially dealing with each segment. The total computation time could be extremely reduced by parallel processing and GPU computing for FFT with sufficient computing power. These details of computation time are now added to the end of Supplementary section V.

Reviewer #2

High resolution, in vivo, deep tissue imaging is very important but also challenging. This paper proposed a new experimental configuration to achieve higher image acquisition rate compared to their previous SLM-based method. Here are my comments:

1. The authors need to explain more details about how to distinguish the multiple scattering and aberration. What are the differences between the processing? Did the authors extract the single scattering signal first and then apply the aberration correction? In the case when aberrations present, the PSF of a single scatterer would also spread out on the detector plane. Also, how to suppress the interruption of speckles?

In our original manuscript, we mainly focused on explaining the image processing of synchronously angular-scanned complex field maps (Methods section and Supplementary section V), and the working principle of the algorithm was referred to our previous works in refs. 28 and 29 (Refs. 31 and 32 in the revised manuscript). Although AO-SASM has new important image processing steps such as the conversion of basis from rotating frame to the lab frame and the correction of local aberration, its working mechanism dealing with scattering and aberration is shared by these previous works. Here, we briefly describe main concept and added this discussion to Supplementary information section XI.

In essence, our algorithm corrects aberration and suppresses speckle noise at the same time. We coherently superpose multiple images taken at different illumination angles. When there exist aberrations, this coherent addition is not fully constructive for the single scattering signal, and the intensity of superposed image is reduced as a consequence. Therefore, we identify the angle-dependent phase correction factors to maximize the total intensity of the reconstructed image. Note that this optimization of total intensity preferably affects to the single scattering. If multiple scattering produces random speckles, their total intensity in the view field doesn't vary much by the addition of the aberration-correction phase factors. For this reason, the algorithm used in AO-SASM works even in the presence of strong multiple scattering noise.

Here is another way to look at our algorithm. Coherent addition of multiple angular images by the aperture synthesis is equivalent to confocal imaging. In case there is no aberration, our coherent addition matches the phase of various illumination angles at any given image pixel, which corresponds to a focused illumination. And we picked up a signal at the corresponding image pixel, which corresponds to sampling through a pinhole. Therefore, coherent aperture synthesis is equivalent to a wide-field confocal imaging. For the case of confocal microscopy, aberrations cause the blurring in both illumination and collection beam paths. This leads to the reduced signal in the confocal image. The same is true for our imaging as aberrations undermine proper accumulation of various angular waves on the way in and out. Since we have recorded a time-gated reflection matrix connecting the relation between all the incident angles and backscattering angles, we can computationally compensate the phase retardations induced by the sample.

Our adaptive optical correction algorithm is based on enhancing the total intensity of the coherent aperture synthesis image. Mathematically, the coherent aperture-synthesis process can be described by the following equation:

$$\begin{aligned} \mathcal{E}_{Coherent}(\Delta\vec{k}) &= \sum_{\vec{k}^i = \vec{k}^o - \Delta\vec{k}} E_{sub}(\vec{k}^o; \vec{k}^i, \tau_0) \\ &= \sqrt{\gamma} \mathcal{O}(\Delta\vec{k}) \cdot \sum_{\vec{k}^i} P_i^a(\vec{k}^i) P_o^a(\vec{k}^i + \Delta\vec{k}) + \sqrt{\beta} \sum_{\vec{k}^i} \mathcal{E}_o^M(\vec{k}^i + \Delta\vec{k}). \end{aligned} \quad (R1)$$

Here, among the matrix elements $E_{sub}(\vec{k}^o; \vec{k}^i, \tau_0)$, those elements whose momentum

difference $\vec{k}^o - \vec{k}^i$ is equal to $\Delta\vec{k}$ are added in their electric fields. Then, the single-scattered waves within $E_{sub}(\vec{k}^o; \vec{k}^i, \tau_0)$ are proportional to the object function $\mathcal{O}(\Delta\vec{k})$ while multiple-scattered waves $\mathcal{E}_o^M(\vec{k}^i + \Delta\vec{k})$ within $E_{sub}(\vec{k}^o; \vec{k}^i, \tau_0)$ are just random realizations of speckle fields varying with \vec{k}^i . Single-scattered wave experiences aberrations on the way in depending on \vec{k}^i and on the way out with $\vec{k}^o = \vec{k}^i + \Delta\vec{k}$. These effects of round-trip aberrations are described by the complex-valued pupil functions $P_i^a(\vec{k}^i)$ and $P_o^a(\vec{k}^i + \Delta\vec{k})$. The γ and β are the intensity of single- and multiple-scattered waves, respectively, and $\beta > \gamma$ in the deep-tissue imaging. The term, $\sum_{\vec{k}^i} P_i^a(\vec{k}^i) P_o^a(\vec{k}^i + \Delta\vec{k})$, significantly undermines the accumulation of the single scattering signal and distorts the point-spread-function. This is because the cross-correlation of the complex-valued pupil functions is always smaller than that of the real-valued pupil functions in the aberration-free case (Eq. (R2))

$$\left| \sum_{\vec{k}^i} P_i^a(\vec{k}^i) P_o^a(\vec{k}^i + \Delta\vec{k}) \right| \leq \left| \sum_{\vec{k}^i} P(\vec{k}^i) P(\vec{k}^i + \Delta\vec{k}) \right|. \quad (\text{R2})$$

Here $P(\vec{k}^i)$ is the ideal aberration-free pupil function, which is unity within the numerical aperture and 0 otherwise. Therefore, the intensity of coherent image $|\mathcal{E}_{Coherent}(\Delta\vec{k})|^2$ would be maximum if there is no aberration. The key idea of our algorithm is to introduce the angle-dependent phase corrections, $\theta_i(\vec{k}^i)$ and $\theta_o(\vec{k}^i + \Delta\vec{k})$ in such a way to maximize the resulting intensity of the coherent image, $\mathcal{E}_{Coherent}^{cor}(\Delta\vec{k})$:

$$\begin{aligned} \mathcal{E}_{Coherent}^{cor}(\Delta\vec{k}) &= \sqrt{\gamma} \mathcal{O}(\Delta\vec{k}) \cdot \sum_{\vec{k}^i} e^{-i\theta_i(\vec{k}^i)} P_i^a(\vec{k}^i) P_o^a(\vec{k}^i + \Delta\vec{k}) e^{-i\theta_o(\vec{k}^i + \Delta\vec{k})} \\ &+ \sqrt{\beta} \sum_{\vec{k}^i} e^{-i\theta_i(\vec{k}^i)} \mathcal{E}_o^M(\vec{k}^i + \Delta\vec{k}) e^{-i\theta_o(\vec{k}^i + \Delta\vec{k})}. \end{aligned} \quad (\text{R3})$$

This will ensure the conversion of complex-valued aberrated pupil functions to ideal real-valued pupil functions. As described in the Methods section, we employed iterative optimization for identifying $\theta_i(\vec{k}^i)$ and $\theta_o(\vec{k}^i + \Delta\vec{k})$. It is important to note that mainly the single-scattered waves take part in this process, and the multiple-scattered waves play little role. The maps of multiple-scattered waves taken at different angles of illumination are uncorrelated with respect to one another, and they remained so even after multiplying the phase corrections. Therefore, the maximization of total intensity is almost exclusively due to the aberration correction of the single-scattered waves. In other words, the application of phase corrections, $\theta_i(\vec{k}^i)$ and $\theta_o(\vec{k}^i + \Delta\vec{k})$, to multiple-scattered waves (the second term in Eq. (R3)) causes little change in the total intensity of multiple-scattered waves in the view field.

Figure R1 shows our theoretical analysis of the monitoring the relative intensities of single- and multiple-scattered waves with the increase of iteration number. As the iteration goes on, the total intensity (red dots) is increased. And we verified that the increase of total intensity is mainly due to the increase of the single scattering (blue dots), and multiple scattering stays almost the same during the entire process.

Figure R1. Relative intensities of single- and multiple-scattered waves with respect to the iteration number. Red, blue, black markers correspond to the total intensity of the coherent aperture synthesis image, the intensity of single-scattered waves, and that of the multiple-scattered waves, respectively. Plots were normalized by the initial contribution of single-scattered waves (Ref. 32 in the revised manuscript).

We could observe a similar trend in our experimental data. We performed a new data analysis for the image taken from 12 dpf zebrafish (Fig. R2). Figure R2a is a time-gated reflection image for a normal illumination, and Fig. R2b is the coherently superposed image of 800 angular images with no aberration correction, which corresponds to conventional OCM image. Single-angle time-gated image (Fig. R2a) was mostly dominated by multiple scattering noise, and it is hard to conceive whether there is any structure or not. Coherently superposed image is better than single-angle image as the existence of myelinated axons can be conjectured. However, the broadening of PSF makes single scattering still smaller than multiple scattering noise. Figures R2c and d show the images after forward and backward optimization processes of the first iteration, respectively, and Fig. R2e was acquired after 5 iterations. We could observe that the ratio of single scattering signal to multiple scattering noise was greatly improved with the increase of iteration number. For the quantitative analysis, we calculated average multiple scattering intensity per pixel over the area indicated by the square box in Fig. R2b where there were no myelinated axons (black dots in Figs. R2f and g). And we computed the average intensity per pixel along the lines where myelinated axons were located. Red dots in Fig. R2f were derived from the area indicated by a white ellipse in Fig. R2b, and those in R2g from the yellow ellipse in Fig. R2b. In each case, we obtained average single scattering intensity per pixel by computing the difference between the intensity at the myelinated axons and that in the background (blue dots in Figs. R2f and g). In accordance with the theory and numerical analysis, we could confirm that the increase of the total intensity by the iterative optimization was mainly due to the increase of single scattering intensity. In Fig. R2h, we show the amplitude profile along the white line in Fig. R2c, where we could observe that the background noise level stays almost the same, and signal was increased only at the myelinated axons.

This new analysis on the intensity changes of single- and multiple-scattered waves with respect to the iteration number is added to the Supplementary section XI.

Figure R2. Image analysis with the increase of iteration number. **a**, Amplitude map of a complex-field map acquired for the normal illumination. **b**, Amplitude map after coherent superposition of complex-field maps taken for 800 different illuminations. Aberration correction was not yet applied. **c** and **d**, Amplitude maps after forward and backward optimizations of the first iteration, respectively. **e**, Amplitude map after 5 iterations. **f**, Average intensity per pixel within the white ellipse in **b** (red dots) and that in the square box in **b** (black dots). Their difference is plotted in blue dots, which corresponds to single scattering signal intensity. **g**, Same as **f**, but within the yellow ellipse in **b**. **h**, Amplitude along the white line in **c** with the increase of iteration number. Scale bar in **a**: 5 μm .

2. From lines 69-71, the authors claimed that most of the developed software-based AO algorithms required guiding stars, which is not correct. Examples can be found from *Opt. Express* 21(9), 10850 (2013), *Opt. Lett.* 41(14), 3324–3327 (2016), *Sci. Rep.* 6, 35209 (2016)

The reviewer’s comment is based only on one sentence in the entire paragraph. The related paragraph in the original manuscript is reiterated below.

“However, most of the developed algorithms require point-like structures that can serve as guiding stars because of their inability to distinguish the aberrations on the way to the specimens from those on the way out. Alternatively, an illumination beam with narrow angular divergence was used to minimize the aberration on the way in, and image metrics such as image sharpness and intensity were optimized by adding corrective phases to the Fourier-transformed map of the acquired image (*Opt. Express* 21(9), 10850 (2013), *Opt. Lett.* 41(14), 3324–3327 (2016), *Sci. Rep.* 6, 35209 (2016)). Because these approaches cannot easily distinguish signals from multiple-scattering noise, they are susceptible to the multiple-scattering noise.”

In the sentence starting from ‘Alternatively,’ we introduced methods that didn’t rely on the guiding stars. In fact, the sentence refers to the references that reviewer suggested. We acknowledge that these references were missing in the original manuscript, and they are now added to the revised manuscript.

We stated that these software-based AOs are susceptible to the multiple-scattering noise. Here, we conducted additional analysis to support this claim. The studies introduced in *Opt. Express* 21(9), 10850 (2013) and *Sci. Rep.* 6, 35209 (2016) considered full-field OCT, which acquires depth image for normal illumination. Therefore, their image is equivalent to our single time-gated complex-field map taken for normal illumination. Due to the use of only single-angle

illumination in these references, the volumetric imaging rate is extremely high. Single-angle illumination also helps to eliminate the effect of aberrations in the illumination beam path. The drawback is that it is susceptible to the multiple-scattering noise because confocal gating is missing. Therefore, finding aberration using the sharpness metric is highly affected by the multiple-scattering noise.

In Fig. R3a. we showed a complex-field map of a resolution target taken for the normal illumination in the absence of aberration and scattering medium. The image was blurred because we numerically defocused the image by $-10\ \mu\text{m}$. We then applied the aberration correction of this defocused image by the method introduced in Sci. Rep. 6, 35209 (2016). This is done by varying the coefficient of the Zernike polynomial related to the defocus aberration, which is equivalent to applying a computational refocusing, and monitoring the image sharpness metric based on Shannon entropy. We readily observed that the image sharpness metric was maximum at the refocusing distance of $10\ \mu\text{m}$ (Fig. R3c), which matches with the expectation. And the reconstructed image after $10\ \mu\text{m}$ refocusing shows a sharp image (Fig. R3b). Therefore, the algorithm works well in the absence of multiple scattering noise.

Figure R3. Comparison of aberration correction performance with software-based approach using full-field OCT with respect to multiple scattering. **a**, Amplitude map of a Siemens star pattern numerically defocused by $-10\ \mu\text{m}$ for the normal illumination. Scale bar: $5\ \mu\text{m}$. **b**, Amplitude map after aberration correction by image sharpness metric. **c**, Variation of image metric with the scanning of the coefficient of the Zernike polynomial corresponding to defocusing. **d**, *En face* image under the hindbrain of a 12 dpf zebrafish numerically defocused by $-10\ \mu\text{m}$ for the normal illumination. **e**, *En face* image after aberrations correction by image sharpness metric. **f**, Variation of image metric with the coefficient of the Zernike polynomial corresponding to defocusing. **g**, Coherently accumulated image of 800 angular scanned complex-field maps intentionally defocused by $-10\ \mu\text{m}$. **h**, AO-SASM image at the same position of the zebrafish. **i**, Aberration map determined by our algorithm. Color bars: phase in radians.

In the presence of multiple scattering noise, the performance of the algorithm is vulnerable. Figure R3d shows the complex-field map of a zebrafish hindbrain taken for the normal

illumination. Due to the dominance of multiple scattering, myelinated axons were almost invisible. We eliminated the sample-induced aberrations using the aberration map identified by the AO-SASM and intentionally added defocus aberration by $-10\ \mu\text{m}$. In a similar way to Fig. 3c, we scanned the coefficient of the fourth-order Zernike polynomials to refocus the image and calculated the sharpness metric. Unlike the case of the resolution target with no scattering medium, the coefficient at which the sharpness metric was maximum was off from the intentional defocus. This is because the sharpness metric was mostly governed by the multiple scattering. To verify the performance of aberration correction of AO-SASM in this multiple-scattering regime, the time-gated reflection matrix was analyzed in Figs. R3g-i. Figure R3g shows the coherently accumulated image of 800 angular scanned complex-field maps intentionally defocused by $-10\ \mu\text{m}$, which obviously couldn't visualize any clear features. However, aberration correction of AO-SASM recovered the clear image of axon fibers by faithfully identifying the intentionally imposed defocus aberration (Figs. R3i). This result proves that aberration correction of AO-SASM is robust to multiple-scattering owing to the coherent accumulation with angularly scanned images.

The work introduced in *Opt. Lett.* 41(14), 3324–3327 (2016) is somewhat similar with our proposed method in the sense that it starts with confocal gating as well as coherence gating. In the OCT and OCM imaging, focused illumination and pinhole detection form a confocal gating. In this reference work, the complex-field map of OCM image was the starting point, which is the same as our coherently superposed image prior to the aberration correction. The aberration correction map based on Zernike polynomials was applied to the Fourier transformed angular spectrum map of the OCM complex-field map in such a way to maximize the image sharpness metric defined by the summation of intensity squared in the image. This algorithm has a major limitation in the case of high-resolution imaging. The Fourier transformed image contains aberrations for both illumination and collection beam paths. Therefore, applying correction to this image cannot deal with the aberration of illumination separately from that in the collection beam path. This may not be a significant issue in the case of low numerical-aperture (NA) imaging, but in high NA imaging the aberration in the illumination beam path causes significant broadening of the illumination PSF.

To elucidate this point, we eliminated both input and output aberrations in the measured time-gated reflection matrix. We then added defocus aberration by $-10\ \mu\text{m}$ to both input and output paths. Figure R4a shows the resulting angular scanned coherently accumulated image of a resolution target before the application of aberration correction, which is equivalent to conventional OCM image. We then scanned the coefficient of the fourth-order Zernike polynomial applied to the Fourier transformed complex-field map of the OCM image. Figure R4c shows the image metric with the scan of the computational refocusing distance using the same method as the reference papers. Two representative image metrics, summation of intensity square and Shannon entropy, were estimated, but there were no perceivable peaks at $10\ \mu\text{m}$. And the computational refocusing back to $10\ \mu\text{m}$ didn't show a clear image (Fig. R4b). This is because the aberration in the illumination path was not dealt with in this reference work. On the other hand, AO-SASM found the aberration maps for both illumination and collection paths (Figs. R4e and f), which correspond to $10\ \mu\text{m}$ defocus, and showed clear object image (Fig. R4d). The main difference of AO-SASM from conventional pinhole-gated OCM is that AO-SASM collects signals at both the point of illumination and the surrounding area. This ultimately enables us to deal with the input and output aberrations separately, thereby enabling high-order aberration correction for high-NA imaging.

Figure R4. Comparison of aberration correction performance with software-based approach using optical coherence microscopy for high-NA imaging. a, Amplitude map numerically defocused by $-10 \mu\text{m}$ for OCM image of a Siemens star pattern. Scale bar: $5 \mu\text{m}$. **b,** Amplitude map after applying $10 \mu\text{m}$ refocusing to the angular spectrum of **a**. **c,** Variation of image metric with the scanning of the coefficient of the Zernike polynomial corresponding to defocusing. **d,** AO-SASM image for the same Siemens star pattern in **a** after applying our algorithm. **e** and **f,** aberration maps for input (**e**) and output (**f**) paths. Color bars: phase in radians.

3. From lines 134-136, “The typical bandwidth used in the experiment was 15 nm . This corresponds to the temporal coherence time of approximately 100 fs , which is the effective width of the pulse front.” First of all, the direct output from the supercontinuum generation is not possible at the femtosecond level (the author didn’t define “fs” as well) because of the chirping. The authors should use the coherence/time gate here instead. The axial resolution should be provided. Secondly, the output bandwidth of the light source could be much larger than 15 nm ? What was the reason to choose only 15 nm ? Larger bandwidth provides better axial resolution. Is there any experimental limitation for using a larger bandwidth?

In fact, the text was already written in such a way that 100 fs is the temporal coherence time, not the absolute pulse width. This corresponds to the coherence length of $30 \mu\text{m}$, which is translated into the depth gating of $15 \mu\text{m}$ after accounting for the epi-detection geometry. And depth resolution set by the confocal gating was measured to be $2.0 \mu\text{m}$ (Fig. R5a), which is close to the theoretical resolution given by $\frac{0.88\lambda}{(n-\sqrt{n^2-NA^2})} \sim 2.0 \mu\text{m}$ (Pawley, J., Handbook of Biological Confocal Microscopy, Springer, 3rd Edition (2006)). This axial resolution by the confocal gating is much shorter than coherence gating due to the use of 0.8 NA objective lens. The axial intensity profile shown in Fig. R5a was measured for a reflecting surface. Its FWHM was $1.7 \mu\text{m}$, and the distance from the center to the first minima was measured to be $2.0 \mu\text{m}$.

The bandwidth of 15 nm was chosen for a reasonable temporal gating to reject multiple scattering from other depths and also for a volumetric imaging within the $15 \mu\text{m}$ -thick coherence volume. AO-SASM acquires wide-field coherent images, which enables 3D imaging within the coherence volume by the axial resolution of the objective lens. In other words,

multiple depth images can be acquired over the range of 15 μm by the axial resolution of 2.0 μm . This can be done by computationally propagating each complex-field image taken at an original depth to a desired depth within the coherence volume. Figures R5b-d shows an example of computational refocusing to different depths from the data acquired at an initial depth.

Figure R5. Axial resolution of the system and computational refocusing to different depths within coherence volume. **a**, Intensity profile along the depth measured for a reflecting surface with 800 angular incident waves. **b-d**, Computational refocusing within a single coherent volume. **c**: image at the original focus. **b** and **d**: images after numerical propagation by $-5 \mu\text{m}$ and $5 \mu\text{m}$ from the original focus. Scale bar, $5 \mu\text{m}$.

For the best axial resolution and imaging depth, the bandwidth of laser should be enlarged to 15 nm or larger such that coherence gating is comparable to the confocal gating. However, this is subject to the system dispersion, which is especially difficult to eliminate for wide-field interferometric imaging with rotating illuminations. And the number of depth images required to cover the sample volume becomes impractically large.

We revised the text as follows to incorporate this discussion on the choice of laser bandwidth.

“The typical bandwidth used in the experiment was 15 nm. This corresponds to the temporal coherence time of approximately 100 fs, which is the effective width of the pulse front, and coherent gating of 15 μm . An objective lens with a numerical aperture (NA) of 0.8 was used to deliver the planar illumination to the sample and capture backscattered waves. Therefore, single-depth recording of complex-field images can cover a depth range of 15 μm at the axial resolution of 2.0 μm set by confocal gating with the use of computational refocusing.”

And we added this discussion on the axial resolution and numerical refocusing to Supplementary section XII.

4. When scanning the different angles of illumination, for each recording, the back-scattering light came from different volumes and different angles related to the camera plane. How to make sure the synthetic information from the same sample plane along depth when imaging a continuous tissue sample? As the reference beam was also scanning, there was no constant reference plane (not the referenced beam) at a specific depth.

This is a critical question since we were also initially unsure whether the simultaneous rotation of sample and reference waves would pick up an image at a specific depth. But it became evident when we looked into the layout carefully. The sampling depth within the tissue is determined by the position of the reference mirror. The tilted waves at the reference mirror and at the corresponding depth in the sample are perfectly synchronized. In the case of backscattering signals at the other depths in the sample, there are phase mismatches that cause

the reconstructed image to be blurred. Note that this synchronous sampling works only for single-scattered waves. Multiple-scattered waves having the same flight time as the reference wave can originate from depths other than that set by the reference mirror such that they are not synchronous with the reference wave. Therefore, single-scattered waves in the time-gated complex-field maps acquired by synchronous angular scanning has the same depth range allowed by coherence gating for all scanning angles.

5. How is the performance of rejecting multiple scattering by comparing this matrix method to a physical pinhole (e.g. fiber) in point-scanning OCT? Although the authors claimed that “where the signals that used to be spread out to off-diagonal elements were gathered together and concentrated on the diagonal elements. This differs from other software-based AO microscopy methods based on the confocal detection, in which the aberration correction does not gather the defocused energy back to the focus...” This sentence was only discussing the aberrations. The example of confocal imaging was not enough as it didn’t provide the coherence gate. I suspect the dynamic range would be decreased compared to common point-scanning FD-OCT as all the useless light from other planes would enter the camera. What is the SNR for the system?

In the absence of aberrations, the multiple scattering rejection capability of our matrix method is equivalent to a point-scanning OCT using a physical pinhole. This is because the coherent addition of multiple angular images based on aperture synthesis is the exact equivalence of the confocal gating. Since all our angular images were taken by the coherence gating, our method is equivalent to the time-domain OCM. However, our matrix method can deal with input and output aberrations separately in such a way that optimal confocal gating is maintained. And the confocal reflectance image was presented in Fig. 3g to provide a reference to earlier zebrafish studies.

Comparison of the dynamic range between AO-SASM and common point-scanning FD-OCT requires a few considerations. Reviewer’s question for the SNR of the system is a good starting point. Let us clarify how the SNR of AO-SASM is determined. At first, we measured the SNR of single-angle complex-field map. Figure R6a shows the intensity of the complex-field map for the normal illumination as a function of the position of the reference mirror. The blue dots were acquired when there was a reflector at the sample plane, and the red dots were obtained when there was no reflector at the sample plane. The signal to background ratio (SBR) defined by the ratio of signal intensity to background intensity was measured to be about 39.7 dB. Signal to noise ratio (SNR), defined by the ratio of signal intensity to the standard deviation of background intensity, was measured to be 39.4 dB.

In AO-SASM, complex-field maps taken for different illumination angles are coherently superposed. Then, both the SBR and SNR increase in proportion to the number of illumination angles, N_{in} (Ref. 31 in the revised manuscript). Figures R6c and d show the AO-SASM images with and without a reflector at the sample plane in the case of $N_{in} = 3,000$. The SBR in this case was measured to be about 74.0 dB, about 30 dB higher than the single-angle imaging as expected. And SNR was measured to be about 74.0 dB, the same as SBR. The background noise was measured by the standard deviation of the image in Fig. R6d. Figure R6b shows the SBR and SNR as a function of N_{in} .

The SNR of our system is about 20-30 dB lower than the point-scanning FD-OCT partly because our detection scheme is in the temporal domain. Typically, the dynamic range of the spectral domain approach is about 20 dB higher than the temporal domain approach. Another reason is the effect of specular reflections from various optical elements. Our system is wide-field detection with high numerical aperture, and it is even equipped with the synchronous angular scanning setup. Therefore, it is likely that unwanted reflections can contribute to the

noise. In fact, bright spots in Fig. R6d are due to the artefact. Considering all these effects, we think that 74 dB SNR is a reasonable level. In particular, with this SNR, we could measure single scattering signal when the intensity of multiple scattering is 74 dB higher.

We added this new data analysis on the system SNR to the Supplementary section XIII.

Figure R6. Sensitivity and SNR of the system. **a**, Intensity profile along the depth for normal incident wave. Blue circles for the reflecting surface, red circles for the case of no sample placed at the sample stage. **b**, SBR and SNR vs the number of angular basis. **c** and **d**, Coherently accumulated images of 3,000 incident angular basis for the cases with a reflecting surface (c) and without the reflecting surface (d). Scale bar, 5 μm .

6. As off-axis geometry was used, the authors should also explain that band-pass filter in the Fourier-domain was used before all the other processing.

We missed this detail in the original manuscript. AO-SASM measures the interferogram with respect to the rotating reference wave in the off-axis interferometric configuration. Therefore, delicate procedure of image processing, including Hilbert transform and coordinate transform from the rotating reference frame to the laboratory frame, is required to construct the time-gated reflection matrix and correct the wavefront aberration. Following the reviewer's suggestion, we described the process of Hilbert transform used for obtaining complex-field map out of raw interferogram in the Supplementary section III as follows.

“In the camera, the measured interferogram in AO-SASM is given by $I(\vec{r}_o; \vec{k}^i, \tau_0) = |E_S(\vec{r}_o; \vec{k}^i, \tau_0) + E_R(\vec{r}_o)|^2 = E_S E_S^* + E_R E_R^* + E_S(\vec{r}_o; \vec{k}^i, \tau_0) E_R(\vec{r}_o)^* + E_S(\vec{r}_o; \vec{k}^i, \tau_0)^* E_R(\vec{r}_o) = I_S + I_R + E_S E_R^* + E_S^* E_R$. Figure S3-1a and Fig. S3-1e show the interferograms of zebrafish for normal incidence and $k^i/k_0=0.8$ respectively. By taking the Fourier transform of $I(\vec{r}_o; \vec{k}^i, \tau_0)$,

angular spectrum $\tilde{I}(\vec{k}; \vec{k}^i, \tau_0) = \tilde{I}_S(\vec{k}) + \tilde{I}_R(\vec{k}) + \widetilde{E_S E_R^*}(\vec{k} - \vec{k}^i - \vec{k}_{DG}) + \widetilde{E_S^* E_R}(\vec{k} + \vec{k}^i + \vec{k}_{DG})$ was obtained (Fig. S3-1b, f). Then we obtained angular spectrum of complex field map in rotating frame $\widetilde{E_S E_R^*}(\vec{k} - \vec{k}^i)$ (Fig. S3-1 c, g) by shifting the frequency of \vec{k}_{DG} and filtering the DC frequency. By taking the inverse Fourier transform of $\widetilde{E_S E_R^*}(\vec{k} - \vec{k}^i)$, complex field map $E_{GM}(\vec{r}_0; \vec{k}^i, \tau_0)$ was obtained (Fig. S3-1d, h), which was then converted to the map in the laboratory frame for the application of aberration correction algorithm.”

7. As there was no physical pinhole, how deep this technique can image? How about comparing it to a common FD-OCT/OCM with hardware-/software-based AO?

Since AO-SASM uses aperture synthesis of time-gated complex field maps, it is equivalent to optical coherence microscopy. Therefore, its imaging depth is comparable to OCM in the absence of aberration. However, our AO-SASM can deal with strong aberrations in the illumination and collection paths separately. As such, it enables us to obtain ideal imaging performance even in the presence of aberrations. According to our previous study, we could achieve the imaging depth of 7 times scattering mean free paths even when there exist significant sample-induced aberrations (Refs. 32 in the revised manuscript). As we explained in detail in response to the reviewer’s comment #2, common FD-OCT/OCM with software-based AOs are either susceptible to multiple scattering noise or unable to separately correct input and output aberrations.

The comparison with hardware-based AO has many aspects to discuss. General aspects were well described in the review papers (Booth, M.J. *Light: Science & Applications* **3**(2014)). Hardware-based AOs typically require multiple image acquisitions, which makes the overall aberration correction process time-consuming. The benefit is to obtain corrected image right after the completion of optimization process. And it can be applied to inelastic scattering processes such as fluorescence imaging modalities. On the other hand, software-based AOs such as the references suggested by the reviewer and our AO-SASM can only be applicable to elastic scattering processes, but there is ample room to deal with aberrations up to high-order after the completion of image acquisition. Therefore, it is suitable for imaging at either highly dynamic conditions such as human retina imaging or strongly aberrating samples required to correct a large number of modes.

8. What was the volumetric rate? It seems this technique is still quite slow, and the in vivo sample has to be fixed.

In our demonstration of *in vivo* imaging, zebrafishes were not fixed, but anesthetized.

Volumetric imaging rate depends on the number of incidence angles N_{in} and camera frame rate determined by the view field. The depth sampling range set by the coherence gating was 15 μm , and axial resolution set by the confocal parameter was 2.0 μm . Therefore, there are at least 7 successive optical sections within a single coherence volume, and multiple *en face* images can be retrieved along the axial direction by the computational refocusing. In our system, the maximum camera frame rate was 400 frames/s for the view field of $22 \times 22 \mu\text{m}^2$. If N_{in} is 100 for the image reconstruction, then we could obtain images over the volume of $22 \times 22 \times 15 \mu\text{m}^3$ in 0.25 seconds. N_{in} should be increased if multiple scattering and sample aberration are more pronounced. Our camera’s maximum frame rate was 60 fps for the view field of $110 \times 110 \mu\text{m}^2$. This corresponds to the volume rate of $100 \times 100 \times 15 \mu\text{m}^3$ per 1.67 seconds in the case of $N_{in} = 100$. In the near future, we are planning to use a high-speed camera to speed up the volumetric imaging rate.

We acknowledge that AO-SASM is intrinsically slower than FD-OCT/OCM software-based AOs that rely on either single-angle illumination imaging or confocal measurements. This is simply because we take more data than the previous techniques. In essence, our time-resolved reflection matrix measurements take multiple angular images, not just a normal illumination image when compared to the full-field OCT. Or reflection matrix is equivalent to a focused illumination and wide-field detection, not a point detection, when compared to OCM. In return, AO-SASM enabled us to address more pronounced aberrations and multiple scattering noise than before. Therefore, direct comparison of the imaging speed is not relevant. The previous approaches are optimal for applications where high-speed imaging is required such as retinal imaging. On the other hand, applications where aberrations and multiple scattering noise are too severe for these previous approaches to handle, then our method can be employed to visualize the structures that are otherwise invisible. We expect that a complementary approach will be useful where conventional high-speed imaging is used for covering wide area and our method is used to interrogate the suspicious/interesting area in depth.

We added the volumetric imaging rate of AO-SASM to the end of Supplementary section II.

9. Because there was no actual wavefront modulator used, the title of AO might cause some confusion. I would suggest changing it to computational or digital AO.

This is a good suggestion. However, the title is already too long to add an additional word. Instead, we put more emphasis on the computational aspect of our technique in the abstract and main text.

Reviewer #3

This paper presents a label-free method for in vivo neural imaging in the larval zebrafish brain. The method is based on a synchronous angle scanning approach, known by the acronym 'CASS' that has previously been presented by the authors (Ref. 28). The compensation of aberrations in combination with CASS has also been demonstrated (Ref. 29). Thus the main novelty of this paper is the application these methods to in vivo neural imaging in the larval zebrafish brain.

The synchronous angular scanning method is presented for the first time in the current paper, and it has never been reported elsewhere. In our previous works in refs. 28 and 29 (Refs. 31 and 32 in the revised manuscript), we fixed the reference wave and used a liquid-crystal spatial light modulator to scan the angle of illumination. While the use of an SLM makes it easy to realize the concept, the image acquisition speed was too slow to record a time-gated reflection matrix of living specimens *in vivo*. With this new development of synchronous angular scanning method, the image acquisition speed is improved by more than 100 times, and the camera frame rate is the limiting factor, not the angular scanning rate. The concept of synchronous angular scanning is not straightforward at all, and it needs careful consideration and special image processing steps. Therefore, the proposed method warrants conceptual novelty of the proposed technique as well as its imaging results.

Since the authors have previously published on the technical aspects of CASS with aberration correction, the significance of the current work should be evaluated with respect to the state-of-the-art imaging approaches for in vivo imaging in the zebrafish and other animal brain models. Relevant papers based on lightsheet, multiphoton microscopy, and other novel confocal approaches such as SCAPE should be cited, and the advantages/disadvantages of the current method should be discussed in the paper. In particular, how do the capabilities of the method compare to the volumetric imaging speed, imaging depth, and overall information content of previous methods, including the OCT work that the authors have already referenced in the paper? For example, Prevedel et al., Nature Methods, 11:7, 2014 and Bouchard et al, Nature Photonics, Vol 9, 2015 have demonstrated volumetric imaging with 20 Hz update rate and faster. Three-photon microscopy has demonstrated deep imaging in mouse brain with cellular resolution (Ouzounov et al, Nature Methods, Vol. 14, 2017). Compared to these previous approaches for neuroscience imaging studies, what advantages/disadvantages does the present label-free approach offer?

The reference works that reviewer suggested are invaluable tools in interrogating living specimens, but their main scopes are different from ours. These are complementary techniques, rather than competing techniques, to our AO-SASM. The main feature of AO-SASM is 'label-free' deep-tissue imaging with a diffraction-limit spatial resolution in the condition where conventional imaging modalities lose resolving power due to the pronounced aberration and multiple scattering noise. Specifically, we demonstrated the visualization of fine myelinated axons in the central nervous system in the hindbrain of a matured zebrafish, whose structures are much more complex than mouse brain tissues. In this specimen, strong sample-induced aberrations cause the blurring of an image and reduce single scattering signal below multiple scattering noise level. AO-SASM offered to resolve these problems and revealed myelin structures with the spatial resolution close to the ideal diffraction limit.

The previous works that the reviewer suggested such as SCAPE (Bouchard et al, Nature Photonics, Vol 9, 2015) and light-field microscopy (Prevedel et al., Nature Methods, 11:7, 2014) were specialized in the high-speed volumetric imaging, not the aberration correction. Therefore,

samples that they were interested in are relatively homogenous tissues with weak aberrations. And multi-photon microscopy with long excitation wavelength (Ouzounov et al, Nature Methods, Vol. 14, 2017) is an excellent approach for deep-tissue imaging, but it mainly aims to deal with multiple scattering rather than aberration. Therefore, it is well suited for either applications requiring cellular level of spatial resolution, not the diffraction-limited resolution, or samples with weak aberrations. In fact, our AO-SASM can also employ long-wavelength illumination to take the same advantage of the three-photon microscopy in dealing with multiple scattering, which is one of our future direction.

Aside from the aberration correction capability of AO-SASM, there is a difference in the contrast mechanism of the proposed method and fluorescence imaging modalities suggested by the reviewer. Fluorescence imaging is a formidable tool in life science since it can visualize specific types of cells or molecules. However, special sample preparation is required in most cases, and artifacts such as photobleaching and blinking often undermine the quantification of molecules/structures of interest in the case of longitudinal study. On the other hand, AO-SASM exploits the difference in the intrinsic reflectivity of the target structures from the surrounding area. Therefore, long-term quantitative study can be made possible as shown in refs. 37 and 38 in the revised manuscript, where confocal reflectance microscopy relying on the sample reflectivity as ours was used for lifelong changes in the cortical myelin. Another example is shown in ref. 39 in the revised manuscript, where wavelength-dependent confocal reflectance imaging helps to quantify the diameter of axons much smaller than the diffraction limit. Since our method uses the same source of contrast as confocal microscopy, we can potentially conduct similar study, but at much deeper depth than these previous studies. Main drawback of label-free reflectance imaging is weak target specificity since reflectivity is an integral quantity determined by many factors such as concentrations and linear susceptibilities of all the constituent molecules. But we can train reflectivity to be specific by the comparative imaging with fluorescence imaging, especially in the case of myelinated axons, as shown in refs. 37 and 38 in the revised manuscript.

We cited the references suggested by the reviewer in the revised manuscript and added the following discussion comparing these reference studies with our method to the discussion section.

“In recent years, the application of multi-photon microscopy such as two/three-photon fluorescence microscopy and second/third-harmonic-generation microscopy for studying a zebrafish has drawn much attention due to its deep penetration through scattering tissues and diverse contrast mechanisms (Ouzounov et al, Nature Methods, Vol. 14, 2017, Light: Science & Applications (2018) 7:12). Fluorescence microscopy can visualize specific types of cells or molecules at the expense of special sample preparations, and often times its volumetric imaging rate could be dramatically improved in the form of light-sheet microscopy (Bouchard et al, Nature Photonics, Vol 9, 2015) and light-field microscopy (Prevedel et al., Nature Methods, 11:7, 2014). And second- and third-harmonic-generation microscopy, while high excitation power is required, visualizes morphological context such as collagen fibers, skin cells, blood cells and lipid accumulations without the use of labeling agents. On the other hand, our AO-SASM visualizes myelinated axons and cellular compartments having higher reflectivity than surrounding tissues with the use of elastic backscattering. AO-SASM can not only complement these existing imaging modalities in terms of information content, but also assist them to cope with the sample-induced aberration. The degradation of the spatial resolution of multi-photon microscopy is inevitable due to the strong sample-induced aberrations in the case of zebrafish, and AO-SASM can potentially provide a solution to correct such aberrations.”

A key concern for in vivo CASS imaging is motion of samples, e.g. from heart beat,

respiration, etc. I suspect that what the authors refer to as ‘sample-induced aberrations’ is not just due to sample-induced aberrations in the traditional sense, but also could contain a significant contribution from the in vivo motion of samples. Sample motion will disrupt the phase of the coherently accumulated signal acquired from multiple illumination angles, since the plane wave components contributing to the PSF, such as shown in Fig 2j, are acquired in a multiplexed fashion as a function of time, rather than simultaneously for imaging at the focal plane of a focused beam. I suspect that the larval zebrafish is quite a stable sample, but can the authors provide some numbers to quantify the level of phase instability introduced by this type of sample, and add a discussion on impact of sample motion for in vivo applications?

As the reviewer pointed out, there are phase fluctuations between different angular measurements. However, they can also be computationally corrected by the post processing. Due to the wide-field measurements, there is no phase fluctuation between the image pixels within each image. Therefore, we could compute cross-correlation between angular images in the aperture synthesis process to find out the angle-dependent phase fluctuation. And this angular phase fluctuations and sample-induced aberrations can be clearly separated in our correction algorithm. For better understanding, we described the correction method of the phase fluctuations as follows, which was reported in our previous works in refs. 31 and 32 in the revised manuscript.

We applied a phase correction step to cope with the phase fluctuations during angular scanning. To this end, we constructed the time-gated reflection matrix $\mathcal{E}_o(\vec{k}^o - \vec{k}^i; j)$, which has the individual column corresponding to the object spectrums in the momentum difference for the illumination of j^{th} angular base. Therefore, the summation of the matrix along row direction (i.e. summation of elements with the same $\Delta\vec{k} = \vec{k}^o - \vec{k}^i$) will lead to the coherent accumulation of single-scattered waves. In order to deal with uncontrolled phase shifts, we added additional phase correction factor, ϕ_j , for each angular base. And the set of ϕ_j 's that maximize the total intensity of the coherent summation $\max_{\phi_j} \sum_{\Delta\vec{k}} |\sum_j \mathcal{E}_o(\Delta\vec{k}; j) e^{i\phi_j}|^2$ will correspond to the phase fluctuations. Once this process is done, we multiply $e^{i\phi_j}$ to $\mathcal{E}_o(\vec{k}^o - \vec{k}^i; j)$ to compensate the phase fluctuation. Then, this fluctuation-free matrix is processed by aberration correction algorithm to find the sample-induced aberration.

As the reviewer conjectured, zebrafishes anesthetized by tricaine were relatively stable during the acquisition of multiple angular images. To quantify the temporal instability of the specimen and verify the imaging performance, we examined 800 angular images taken by the frame rate of 400 frames/s. Figure R7a shows the reconstructed image with 800 angular images, and Fig. R7b displays the identified phase fluctuation during angular scanning. There were phase fluctuations possibly due to cardiac impulse and blood flow of the zebrafish, but they were accounted for in the image reconstruction step. We extracted four sub-sets composed of 200 images from 800 images along the sequence with the time interval of 0.5 sec and reconstructed an image with each subset. As shown in Figs. R7e-h and line profiles in Figs. R7c-d, the structures of the myelinated axons were almost stable during one set of measurements.

Figure R7. Temporal stability and AO-SASM images reconstructed by the subset of 800 images taken at 400 frames/s. a, Reconstructed image of the living zebrafish using all 800 images. **b**, Temporal stability quantified by phase fluctuations during angular scanning. **c** and **d**, Line profiles of images in e-h along the blue and green dashed lines in **a**, respectively. **e-h**, Images reconstructed by the first, second, third, and fourth set composed of 200 images along the sequence with the time interval of 0.5 sec, respectively, out of the total 800 images.

Lines 174-175 state that a 21-dpf larval zebrafish introducing both strong aberration and multiple scattering was used in this study. What evidence can the authors provide to support their claim about multiple scattering? Are they able to estimate the imaging in terms of the number of scattering lengths for the results in Fig. 2?

Our aberration correction algorithm enhances the total intensity of the image obtained by coherently accumulating multiple angular images. This leads to the preferential increase of single scattering intensity, which is confirmed by both numerical and experimental data analysis. Specifically, the experimental data analysis shown below reveals the changes of single scattering signal and multiple scattering noise with the increase of iteration number, which clarifies how our algorithm distinguishes the single- and multiple-scattered waves.

We performed a new data analysis for the image taken from 12 dpf zebrafish (Fig. R8). Figure R8a shows a time-gated reflection image taken for a normally incident wave, and Fig. R8b is the coherently superposed image of 800 angular images before applying the aberration correction, which corresponds to conventional OCM image. Single-angle time-gated image was mostly dominated by multiple scattering noise. Coherently superposed image is better than single-angle image to the extent that the existence of myelinated axons can be conjectured. However, the broadening of PSF makes single scattering signal still smaller than or comparable to multiple scattering noise. Figures R8c and d show the images after forward and backward optimization processes of the 1st iteration, respectively, and Fig. R8e was acquired after 5 iterations. We could observe that the ratio of single scattering signal to multiple scattering noise was greatly improved with the increase of iteration number.

For the quantitative analysis, we calculated average multiple scattering intensity per pixel over the area indicated by the rectangular box in Fig. R8b where there is no myelinated axon (black dots in Figs. R8f and g). And we computed the average intensity per pixel along the lines where myelinated axons were located. Red dots in Fig. R8f were derived from the area indicated by a

white ellipse in Fig. R8b, and those in R8g from the yellow ellipse in Fig. R8b. In each case, we obtained the average single scattering intensity per pixel by computing the difference between the intensity at the myelin and that in the background (blue dots in Figs. R8f and g). In accordance with the theory and numerical analysis, we could confirm that the increase of the total intensity by the optimization is mainly due to the increase of single scattering intensity. Figure R8h shows the amplitude profile along the white line shown in Fig. R8c, where we could observe that the background noise level stayed almost the same and signal was increased only at the myelinated axons. A series of these analyses provides the evidence for strong aberration and multiple scattering of the zebrafish.

Figure R8. Image analysis with the increase of iteration number. **a**, Amplitude map of a complex-field map acquired for normal illumination. **b**, Amplitude map of coherently accumulated complex-field maps taken for 800 different illuminations before applying the correction of aberration. **c** and **d**, Amplitude maps after forward and backward optimizations of the first iteration, respectively. **e**, Amplitude map after 5 iterations. **f**, Average intensity per pixel within the white ellipse in **b** (red dots) and that in the rectangular box in **b** (black dots). Their difference is plotted in blue dots. **g**, Same as **f**, but within the yellow ellipse in **b**. **h**, Amplitude profile along the white line in **c** with the increase of iteration number.

This new analysis on the intensity changes of single- and multiple-scattered waves with respect to the iteration number is added to the Supplementary section XI.

Scattering mean free path of biological tissues is a difficult quantity to measure, especially in the case of *in vivo* imaging. To precisely assess the scattering mean free path, we need to measure ballistic photons by varying the thickness of the same type of scattering medium. In the case of zebrafish imaging, scattering property varies from depth to depth because the internal structures change with depth as shown in Figs. 3a-e in the manuscript. One approximate way for estimation is to measure the average intensity of confocal reflectance imaging, which is a measure of ballistic photons, as a function of depth. Since we haven't measured the confocal reflectance imaging for the full depth of a 21 dpf zebrafish, we present here a 10 dpf zebrafish data (Fig. R9a) taken at the wavelength of 633 nm. Figure R9b shows the average intensity as a function of depth for the white dotted rectangular area in Fig. R9a. Figure R9c was acquired from the blue dotted-square area in Fig. R9a. The intensity profiles cannot be fit to a single exponential curve (red curves) partly because confocal reflectance imaging picks up multiple scattering noise at the depths deeper than 50-100 μm and partly because scattering properties

varied with depth. Therefore, we performed exponential curve fittings up to a shallow depth (green curves). The approximate scattering mean free paths were around 50 to 100 μm . This corresponds to the attenuation of single scattering signal by a factor of 10^2 - 10^4 times at the depth of 200 μm in the case of normal-illumination complex-field map, and the sample-induced aberrations further attenuate single scattering intensity. This explains why myelinated axons were invisible in the individual complex-field maps.

Figure R9. Scattering property of a 10-dpf larval zebrafish. **a**, Dorsal view of confocal image encompassing the hindbrain. **b** and **c**, Intensity profiles along the depth in the white dashed rectangle (b) and blue dashed square (c), respectively.

We added this discussion on scattering mean free path to the Supplementary section IX.

The authors state that they have achieved a record effective aberration correction speed of 10,000 modes/s. Can the authors include a justification for this number in the main paper or supplementary information? Since the authors are doing a software aberration correction in post processing, why is the “effective aberration correction speed” that they calculated a fair comparison to traditional AO methods? What was the total computation time required to reconstruct aberration corrected en face images and for the whole reconstructed volume? Also, can the authors provide some evidence that a correction with 10k modes were required (otherwise it is meaningless to try and perform this with traditional AO methods), and therefore why this is a fair comparison to hardware correction methods?

This is a critical question, and the answer to this given below signifies the distinction of our method from the previous computational AO methods.

The effective aberration correction speed of 10,000 modes/s excluding the computation time was already well explained in the supplementary section VI. Maximum frame rate of our camera is 450 Hz for a view field of $22 \times 22 \mu\text{m}^2$, which consists of 400×400 pixels. Since diffraction-limit resolution is 480 nm, there are $N_{max} = 2,285$ angular frequencies in each complex-field map. If we consider the number of incident angles $N_{in} = 100$, it takes $t_m = 0.22$ seconds to complete a set of measurements. Therefore, mode recording speed is given by $\frac{N_{max}}{t_m} = 10300 \approx 10,000$ modes/s since our aberration correction algorithm identifies the angle-dependent phase retardations for all the angular frequencies constituting each complex-field map. The reason we excluded the data processing time in the effective aberration correction speed is that only the data acquisition time is critical for *in vivo* computational adaptive optics imaging unless the real-time visualization is mandatory. Once sufficient number of modes are measured within a data acquisition time short enough to be free from sample perturbation, we can correct sample-induced aberrations all in the post-processing steps. This capability is not available in the hardware-based AO, where aberration correction and focusing should be completed at the time of imaging. On the other hand, hardware-based AO can be

applied to multiphoton microscopy, while AO-SASM itself acquires a coherent backscattering map. Due to these different considerations, the direct comparison of mode correction speed is complicated. We therefore revised sentences in the abstract and main text in such a way to refrain from direct comparison with existing AO methods.

“In doing so, we enhanced the effective aberration correction speed excluding the data processing time to 10,000 modes/s, which is critical for *in vivo* imaging of dynamic specimens with significant sample-induced aberrations.”

“In doing so, we significantly reduced the image acquisition time per depth from a few minutes to 0.22 s and enhanced the effective aberration correction speed excluding the data processing time to 10,000 modes/s (see Supplementary section VI).”

The computation time depends on the field of view and the number of segments. When a personal computer with CPU clock speed of 3.6 GHz was used, it took about 20~30 minutes for reconstructing the coherent volume of $110 \times 110 \times 15 \mu\text{m}^3$. The iterative aberration correction for a single segment of $22 \times 22 \mu\text{m}^2$ took about an order of ten seconds. It took about tens of minutes to calculate *En face* image of the full field of $110 \times 110 \mu\text{m}^2$, which is composed of 6×6 segments. In addition, it took about ten minutes to computationally propagate the acquired *en face* image to 10 different depths within the coherent volume of $110 \times 110 \times 15 \mu\text{m}^3$. Total computation time for a whole central nerve system encompassing the hindbrain of the zebrafish shown in Fig. 3, which consists of 120 coherent volumes or 43,200 segments, was around 50 hours. However, there are many rooms to optimize the computation since the current algorithm is sequentially dealing with each segment. The total computation time could be extremely reduced by parallel processing and GPU computing for FFT with sufficient computing power. These details of computation time are now added to the end of Supplementary section V.

We analyzed the necessity of correcting all the angular modes in the view field using the data shown in Fig. S6k-o. The first column in Fig. R10a show the aberration map and the reconstructed AO-SASM image. We decomposed the aberration maps into Zernike polynomials. Figure R10b shows amplitudes of various orders of Zernike polynomials constituting the aberration map of AO-SASM. One can notice that there are non-negligible contributions even for the 1000th order of the polynomials. We then reconstructed aberration-corrected image after using only the first 50 orders of polynomials of Fig. R10b. The aberration maps and reconstructed image are shown at the second column in Fig. R10a. The aberration was so severe that structures remained blurred even with the use of 50 orders. And the signal level of the reconstructed image with 50 Zernike modes is much smaller than that of AO-SASM image since the intensity is proportional to the Strehl ratio. The third and fourth columns of Fig. R10a show the results with the use of 100 and 500 orders, respectively. We can observe the gradual increase of image quality, but the sharpness, intensity, and signal to background ratio are still below the level shown by AO-SASM. Specifically, Fig. R10c shows the total intensity of the reconstructed with the increase of the number of Zernike polynomial orders. The total intensity increased up to the use of 175 orders and reaches the saturation level, which is far lower than the total intensity of AO-SASM. Note that conventional software-based AOs seldom correct aberration even up to 100 orders, where the aberration correction is not yet ideal. This tells us that correction of aberrations in the angular basis is critical and advantageous in dealing with the severe aberrations. This is partly because Zernike polynomials cannot recover abrupt variations of aberration in the pupil and partly because angular basis is orthogonal and complete in the electric field. In Fig. R10d, we also plotted the intensity profiles along the dashed line shown in Fig. R10a as a function of the number of polynomial orders. We can notice that only AO-SASM can faithfully recover the fine details of structures. This confirms that the use of full angular modes is critical to obtain spatial resolution close to ideal spatial resolution especially when

aberration in the pupil plane varies pixel-by-pixel level.

We added the above discussion on the benefit of AO-SASM dealing with all the angular modes with respect to previous software-based AOs to the end of Supplementary section X.

Figure R10. Image reconstruction with finite orders of Zernike polynomials. **a**, The first column shows aberration map and reconstructed image in the case of AO-SASM. The second, third and fourth columns correspond to the cases of using 50, 100, and 500 Zernike polynomials. **b**, Decomposition of the aberration identified by AO-SASM into various orders of Zernike polynomials. **c**, Total intensity of reconstructed images in **a** as a function of the number of orders of polynomials. Red line indicates the case of AO-SASM. **d**, Intensity profiles along the dashed line shown in **a** depending on the number of orders of polynomials. Red dots were derived from AO-SASM image.

The authors state that the number of angles N_{in} can be significantly less than the number of modes in the field of view. However, I think that acquiring fewer measurements should have an impact on the information content of the reconstructed image. A fewer number of angles would likely reduce the field of view of the reconstructed image due to a drop of signal at the edges of the field of view. Can the authors add a discussion on the trade-offs involved with acquiring lower number of angles?

Reducing the number of incidence angles, N_{in} , affects to achievable imaging depth, not the field of view. The coherent accumulation of multiple angular images enhances single scattering signal to multiple scattering noise ratio by N_{in} times (Ref. 31 in the revised manuscript). Therefore, image reconstruction by a smaller number of angular images becomes more susceptible to the multiple scattering noise. For example, the system SNR reaches 74 dB when $N_{in}=3,000$ while it is only about 39.4 dB when $N_{in}=1$. This is why conventional software-based AO starting from single complex-field map (Sci. Rep. 6, 35209 (2016)) is prone to the multiple

scattering noise. A good strategy to improve data acquisition speed is to use smaller number of illumination angles at a shallower depth and to increase N_{in} with the increase of depth.

We added this discussion on the effect of reducing N_{in} to the end of Supplementary section VI.

What was the overall volumetric imaging time or rate? Only the en face update rate of 4Hz was reported. Was the depth sampling equal to the coherence length of the source, and so how many planes were acquired along the depth axis?

Volumetric imaging rate depends on N_{in} and the camera frame rate. As the reviewer conjectured, the depth sampling is equal to the coherence length of the source, which was 15 μm in our experiment, and axial resolution was about 2 μm . Therefore, there are at least 7 successive optical sections at axial resolution within a single coherence volume, from which more *en face* images can be retrieved along the axial direction by computational refocusing for better 3D visualization. If 100 angular images are used with 400 frames/s, then we could obtain images over the volume of $22 \times 22 \times 15 \mu\text{m}^3$ in 0.25 seconds. If multiple scattering and aberration are more pronounced, then N_{in} should be increased to ensure enough SNR to resolve the target structures. Our camera frame rate was 60 fps for the view field of $110 \times 110 \mu\text{m}^2$, which corresponds to the volume image rate of $110 \times 110 \times 15 \mu\text{m}^3$ per 1.67 seconds.

We acknowledge that AO-SASM is intrinsically slower than FD-OCT/OCM software-based AOs that rely on either single-angle illumination imaging or confocal measurements. This is simply because we take more data than the previous techniques. In essence, our time-resolved reflection matrix measurements take multiple angular images, not just a normal illumination image when compared to the full-field OCT. Or reflection matrix is equivalent to a focused illumination and wide-field detection, not a point detection when compared to OCM. In return, AO-SASM enabled us to address more pronounced aberrations and multiple scattering noise than before. Therefore, direct comparison of the imaging speed is not relevant. The previous approaches are optimal for applications where high-speed imaging is required such as retinal imaging. On the other hand, applications where aberrations and multiple scattering noise are too severe for these previous approaches to handle, then our method can be employed to visualize the structures that are otherwise invisible. We expect that a complementary approach will be useful where conventional high-speed imaging is used for covering wide area and our method is used to interrogate the suspicious/interesting area in depth.

We added volumetric imaging rate of AO-SASM to the end of Supplementary section II.

Strehl ratio is normally given as a number between 0 and 1, whereas Lines 203-204 a Strehl ratio of 52 is reported. If the authors cannot give the actual Strehl ratio, which compares to the ideal diffraction limited spot, then it may be better to report the factor by which the Strehl ratio improved.

We appreciate the reviewer's pointing out the mistake. Although the sentence made it clear what the factor '52' means, we should have used the enhancement of Strehl ratio rather than Strehl ratio itself. The sentence is now revised as follows.

“The enhancement of the Strehl ratio defined by the ratio of peak intensities after and before the aberration correction was 52 (see Supplementary section V for detailed analysis).”

In our original Supplementary information section V, we estimated the absolute value of Strehl ratio, and it was estimated to be 0.94 after the optimization. To precisely assess the Strehl ratio, the aberration corrected PSF at the final iteration was compared to the ideal PSF of an aberration-free system. The peak value of the ideal PSF is set by unity, and the energy of the

aberration corrected PSF was made equal to the energy of the ideal PSF by assuming that the energy of the PSF is fully optimized. Then the peak height of the corrected PSF was measured to be 0.94, which is the estimated Strehl ratio.

In the abstract the authors say that they were able to visualize “anatomical details including fine neuronal processes”. I would suggest that the authors do not use the phrase “neuronal processes” to describe anatomical structure, since “processes” give the impression of time-lapse monitoring of a biological function.

We agree with the reviewer. In the revised manuscript, we replaced neuronal processes with neural fiber.

In the Supplementary Info: It will be helpful to provide additional details of the optical system in fig. S1, such as focal length/part numbers of lenses, telescope magnifications, etc. Also, line 115 states that the AO algorithm maximizes “total intensity of the reconstructed image”. Doesn’t an aberrated PSF have the same energy as a computationally refocused PSF, just that the energy is blurred out in space? Can the authors clarify if they use sum of pixel intensities in the image or mean peak reconstructed intensity?

Following the reviewer’s suggestion, we added the important details of the optical setup to Fig. S1.

The reviewer asked key questions about the working principle of our algorithm, and here we clarified why our AO-SASM is distinct from other existing algorithms. The answer to the question whether the aberrated PSF has the same energy as a computationally refocused PSF is ‘no’ in AO-SASM. If an adaptive optics is applied to a single-angle illumination image, reviewer’s comment is correct. However, we coherently superpose multiple images taken by different angles. When there exist aberrations, this coherent addition is not fully constructive such that the intensity of superposed image is reduced. Note that this optimization of total intensity affects only to the single scattering. If multiple scattering is well-developed random speckles, their total intensity doesn’t vary by the coherent addition with phase correction factors.

Here is another way to look at our algorithm. Coherent aperture synthesis of multiple angular images is equivalent to confocal image. In case there is no aberration, our coherent addition matches the phase of various illumination angles at any given image pixel, which corresponds to a focused illumination. And we picked up signal at the corresponding image pixel, which is equal to sampling through a pinhole. Therefore, coherent aperture synthesis is equivalent to a wide-field confocal imaging. For the case of confocal microscopy, aberrations cause the blur for both the focused illumination and returning wave such that the single scattering signal picked up by confocal pinhole will be reduced. The same is true for our imaging as aberrations undermine the coherent accumulation of various angular waves in both input and output pathways.

Now the reviewer asked an important next question on whether we use the sum of pixel intensities in the image or mean peak reconstructed intensity. In AO-SASM, we maximized the sum of pixel intensities in the reconstructed image. This is critical especially in the presence of multiple scattering. If we optimize a peak intensity at any given pixel, coherent superposition will likely to induce the constructive interference of multiple scattering rather than single scattering. While multiple scattering can be maximized at any single pixel, its sum of intensity across the entire field of view doesn’t vary much by the control of the addition of angular phase correction factors because speckles are decorrelated from pixels to pixels. On the other hand, intensity sum of single scattering can be increased and maximized as they are coherent across the field of view. This is the main reason why our optimization of total intensity leads to the preferential increase of single scattering.

In the revised manuscript, we clarified that the total intensity means the sum of pixel intensities in the image.

In fact, we have a rigorous theory behind our algorithm (Ref. 32 in the revised manuscript), and here we briefly introduce its essence for the reviewer's reference. Mathematically, the coherent image is expressed by the cross-correlation between the complex pupil functions of the input and output paths:

$$\begin{aligned}\mathcal{E}_{Coherent}(\Delta\vec{k}) &= \sum_{\vec{k}^i=\vec{k}^o-\Delta\vec{k}} E_{Sub}(\vec{k}^o; \vec{k}^i, \tau_0) \\ &= \sqrt{\gamma}\mathcal{O}(\Delta\vec{k}) \cdot \sum_{\vec{k}^i} P_i^a(\vec{k}^i)P_o^a(\vec{k}^i + \Delta\vec{k}) + \sqrt{\beta} \sum_{\vec{k}^i} \mathcal{E}_o^M(\vec{k}^i + \Delta\vec{k}).\end{aligned}\quad (\text{R4})$$

Here, among the matrix elements $E_{Sub}(\vec{k}^o; \vec{k}^i, \tau_0)$, those elements whose momentum difference $\vec{k}^o - \vec{k}^i$ is equal to $\Delta\vec{k}$ are added. Then, the single-scattered waves within $E_{Sub}(\vec{k}^o; \vec{k}^i, \tau_0)$ are proportional to the object function $\mathcal{O}(\Delta\vec{k})$ while multiple-scattered waves $\mathcal{E}_o^M(\vec{k}^i + \Delta\vec{k})$ within $E_{Sub}(\vec{k}^o; \vec{k}^i, \tau_0)$ are just random realizations with \vec{k}^i . Furthermore, single-scattered wave experience aberrations on the way in with \vec{k}^i and on the way out with $\vec{k}^o = \vec{k}^i + \Delta\vec{k}$. These effects of round-trip aberration are described by the complex pupil functions $P_i^a(\vec{k}^i)$ and $P_o^a(\vec{k}^i + \Delta\vec{k})$. And γ and β are the attenuation factors of single- and multiple-scattered waves, respectively. The term, $\sum_{\vec{k}^i} P_i^a(\vec{k}^i)P_o^a(\vec{k}^i + \Delta\vec{k})$, significantly undermine the accumulation of single scattering signal and distorts PSF. This is because the cross-correlation of the complex-valued pupil functions is always smaller than that in the aberration-free case (Eq. (R5)).

$$\left| \sum_{\vec{k}^i} P_i^a(\vec{k}^i)P_o^a(\vec{k}^i + \Delta\vec{k}) \right| \leq \left| \sum_{\vec{k}^i} P(\vec{k}^i)P(\vec{k}^i + \Delta\vec{k}) \right| \quad (\text{R5})$$

Here $P(\vec{k}^i)$ is the ideal aberration-free pupil function, which is unity within the numerical aperture and 0 otherwise. If there were no aberration, the intensity of coherent image $|\mathcal{E}_{Coherent}(\Delta\vec{k})|^2$ would be maximum. The key idea of our algorithm is to apply the angle-dependent phase corrections, $\theta_i(\vec{k}^i)$ and $\theta_o(\vec{k}^i + \Delta\vec{k})$ in such a way to maximize the resulting intensity of the coherent image, $\mathcal{E}_{Coherent}^{cor}(\Delta\vec{k})$:

$$\begin{aligned}\mathcal{E}_{Coherent}^{cor}(\Delta\vec{k}) &= \sqrt{\gamma}\mathcal{O}(\Delta\vec{k}) \cdot \sum_{\vec{k}^i} e^{-i\theta_i(\vec{k}^i)}P_i^a(\vec{k}^i)P_o^a(\vec{k}^i + \Delta\vec{k})e^{-i\theta_o(\vec{k}^i+\Delta\vec{k})} \\ &\quad + \sqrt{\beta} \sum_{\vec{k}^i} e^{-i\theta_i(\vec{k}^i)}\mathcal{E}_o^M(\vec{k}^i + \Delta\vec{k})e^{-i\theta_o(\vec{k}^i+\Delta\vec{k})}.\end{aligned}\quad (\text{R6})$$

This will ensure the conversion of complex-valued pupil functions to ideal real-valued pupil functions. As described in the Methods section, we employed iterative optimization for identifying $\theta_i(\vec{k}^i)$ and $\theta_o(\vec{k}^i + \Delta\vec{k})$. It is important to note that mainly the single-scattered waves take part in this process, and the multiple-scattered waves play little role. The maps of multiple-scattered waves taken at different angles of illumination are uncorrelated with respect to one another remained so even after multiplying the phase corrections. Therefore, the maximization of total intensity is almost exclusively due to the aberration correction of the single-scattered waves. In other words, the application of phase corrections, $\theta_i(\vec{k}^i)$ and

$\theta_0(\vec{k}^i + \Delta\vec{k})$, to multiple-scattered waves (the second term in Eq. (R6)) doesn't change the intensity of multiple-scattered waves by much.

Figure R11. Relative intensities of single- and multiple-scattered waves with respect to the iteration number. Red, blue, black markers correspond to the total intensity of the coherent image, the intensity of single-scattered waves, and that of the multiple-scattered waves, respectively. Plots were normalized by the initial contribution of single-scattered waves.

Figure R11 shows our theoretical analysis monitoring the relative intensities of single- and multiple-scattered waves with the increase of iteration number. As the iteration goes on, the total intensity (red dots) is increased. And the increase of total intensity is mainly due to the increase of the single scattering (blue dots), and multiple scattering stays almost the same during the entire process. We could observe a similar trend in our experimental data as shown in Fig. R8, which is also added to the Supplementary section XI.

Reviewers' comments:

Reviewer #1 (Remarks to the Author):

I am satisfied with the changes made by the authors and their replies in the rebuttal, and I recommend the manuscript for publication.

I would suggest, though now a comparison and discussion of MP results is included, they stress this in the text in the more quantitative/direct way they have done in the rebuttal (cf. At this depth of 237 μm , a single-cell level of spatial resolution was obtained, not the diffraction limit resolution. We conjecture that sample-induced aberration played a role in the degradation of the spatial resolution. On the contrary, AO-SASM reached the depth of 220 μm through a thick hindbrain with the spatial resolution close to the ideal diffraction limit. Therefore, the imaging depth of our method is better than or comparable to the state-of-the-art multi-photon microscopy if spatial resolution is accounted for in estimating the imaging depth.

We would like to stress that it is not straightforward to reach the imaging depth of 200 μm with diffraction-limit spatial resolution in the case of zebrafish due to the severe sample-induced aberration. Unlike mouse brain tissues, many heterogeneous structures such as skin and internal organs in the zebrafish act as sources of aberration. In fact, when we surveyed vast amount of literatures reporting the use of multi-photon microscopy for studying the central nervous system of the zebrafish, the imaging depth was much shallower than 200 μm , and studies were limited to early developmental stages.)

Further, as far as I could see, do not mention the laser power incident on the sample, which is of course relevant for in-vivo application.

Reviewer #2 (Remarks to the Author):

The revised manuscript has clarified many missing details and added more comprehensive calculations, experiments, and discussion. Overall, the manuscript has been improved significantly. My concerns have been well addressed, and I am convinced of this technique. But I believe that the authors should still change the title of AO to computational AO because there was no physical device to "adaptively" correct the aberrations and multiple scattering noise during the experiment. The change of the title can reduce the confusion and emphasize the computational nature of this technique.

Reviewer #3 (Remarks to the Author):

I would like to commend the authors on their comprehensive and thorough response to reviewer comments. In light of these responses and the associated changes the authors have made to the main manuscript and supplementary materials, I recommend publication in Nature Communications after they address the following additional points:

In the added text to address the comparison to other interferometric modalities (Lines 69-75), the authors point to the "inability to distinguish the aberrations on the way to the specimens from those on the way out". In order to clearly establish the benefits of AO-SASM, I think the authors also need to (1) provide evidence why the aberration on the way in vs. on the way out needs to be corrected separately, and (2) when describing the principle of AO-SASM in the main manuscript, clearly state which acquisition step/experimental design/processing step implements the separate correction of aberrations on the way in vs. on the way out.

In Fig. R9 (which is also Fig. S9), and the associated text, the authors state that the approximate scattering mean free paths were around 50 to 100 μm . If the scattering length $l_s = 50 \mu\text{m}$, then at a depth of 200 μm this is only 4 scattering mean free paths, and the stated "attenuation of single scattering signal by a factor of 10^2 - 10^4 times at the depth of 200 μm " seems high. Is

this for double pass detection, and is it referring to attenuation of the amplitude or power of the single scattering signal? Also, for the sake of clarity it would help to separately mention the impact of multiple scattering vs. aberrations. And so I recommend separating the last part of the sentence "and the sample-induced aberrations further attenuate single scattering intensity" into a separate sentence that separately mentions the impact of aberrations.

Reviewers' comments:

Reviewer #1

I am satisfied with the changes made by the authors and their replies in the rebuttal, and I recommend the manuscript for publication.

I would suggest, though now a comparison and discussion of MP results is included, they stress this in the text in the more quantitative/direct way they have done in the rebuttal (cf. At this depth of 237 μm , a single-cell level of spatial resolution was obtained, not the diffraction limit resolution. We conjecture that sample-induced aberration played a role in the degradation of the spatial resolution. On the contrary, AO-SASM reached the depth of 220 μm through a thick hindbrain with the spatial resolution close to the ideal diffraction limit. Therefore, the imaging depth of our method is better than or comparable to the state-of-the-art multi-photon microscopy if spatial resolution is accounted for in estimating the imaging depth.

We would like to stress that it is not straightforward to reach the imaging depth of 200 μm with diffraction-limit spatial resolution in the case of zebrafish due to the severe sample-induced aberration. Unlike mouse brain tissues, many heterogeneous structures such as skin and internal organs in the zebrafish act as sources of aberration. In fact, when we surveyed vast amount of literatures reporting the use of multi-photon microscopy for studying the central nervous system of the zebrafish, the imaging depth was much shallower than 200 μm , and studies were limited to early developmental stages.)

Further, as far as I could see, do not mention the laser power incident on the sample, which is of course relevant for in-vivo application.

We appreciate the reviewer's acknowledgement of our response and providing us with additional valuable comments. Following the reviewer's suggestion, we revised the paragraph in the previous rebuttal letter and added it to the discussion section of the revised manuscript.

“While these multi-photon microscopy techniques provide valuable molecular contrast, they suffer from the loss of spatial resolution due to the sample-induced aberrations especially when imaging zebrafishes. Unlike mouse brain tissues, many heterogeneous structures such as skin and internal organs in the zebrafish act as sources of aberration. According to the literatures reporting the use of multi-photon microscopy for studying the central nervous system of the zebrafish, the imaging depth was much shallower than 200 μm , and studies were limited to early developmental stages. On the contrary, AO-SASM reached the depth of 220 μm through a thick hindbrain with the spatial resolution close to the ideal diffraction limit. Therefore, the imaging depth of our method is better than or comparable to the state-of-the-art multi-photon microscopy if spatial resolution is accounted for in estimating the imaging depth. In this respect, AO-SASM can not only complement these existing imaging modalities in terms of information content, but also assist them to cope with the sample-induced aberration.”

We also added the following sentences to the discussion section to signify the benefit of AO-

SASM in terms of phototoxicity.

“Irradiance of illumination used in AO-SASM was below $100 \mu\text{W}/\text{mm}^2$, which is an order of magnitude lower than maximum permissible exposure level of $200 \text{ mW}/\text{cm}^2$ in animal tissues. It requires relative weak excitation beam in comparison with multi-photon microscopy as elastic backscattering is used for imaging. Furthermore, wide-field detection configuration is used instead of pinhole gating, which makes the signal collection more efficient.”

Reviewer #2

The revised manuscript has clarified many missing details and added more comprehensive calculations, experiments, and discussion. Overall, the manuscript has been improved significantly. My concerns have been well addressed, and I am convinced of this technique. But I believe that the authors should still change the title of AO to computational AO because there was no physical device to “adaptively” correct the aberrations and multiple scattering noise during the experiment. The change of the title can reduce the confusion and emphasize the computational nature of this technique.

We are glad to know the reviewer was convinced by our explanations. Following the reviewer’s suggestion, we changed the title of the manuscript as follows to emphasize the computational nature of aberration correction.

“Label-free neuroimaging *in vivo* using synchronous angular scanning microscopy with single-scattering accumulation algorithm”

Reviewer #3

I would like to commend the authors on their comprehensive and thorough response to reviewer comments. In light of these responses and the associated changes the authors have made to the main manuscript and supplementary materials, I recommend publication in Nature Communications after they address the following additional points:

It is our pleasure to know that our response has clarified the reviewer’s concerns. We addressed the remaining points in the following.

In the added text to address the comparison to other interferometric modalities (Lines 69-75), the authors point to the “inability to distinguish the aberrations on the way to the specimens from those on the way out”. In order to clearly establish the benefits of AO-SASM, I think the authors also need to (1) provide evidence why the aberration on the way in vs. on the way out needs to be corrected separately, and (2) when describing the principle of AO-SASM in the main manuscript, clearly state which acquisition step/experimental design/processing step implements the separate correction of aberrations on the way in vs. on the way out.

As an evidence why the aberration on the way in and out needs to be corrected separately, we provided data analysis of the experimentally acquired data in the previous response letter, which is given below. We compared the performance of aberration correction between ordinary OCM, which treats aberrations on the way in and out as a whole, and AO-SASM. In the case of high-numerical aperture-imaging, we show that only AO-SASM works properly. To make this point clear, we added this discussion to the SI.

The work introduced in Opt. Lett. 41(14), 3324–3327 (2016) is somewhat similar with our proposed method in the sense that it starts with confocal gating as well as coherence gating. In the OCT and OCM imaging, focused illumination and pinhole detection form a confocal gating. In this reference work, the complex-field map of OCM image was the starting point, which is the same as our coherently superposed image prior to the aberration correction. The aberration correction map based on Zernike polynomials was applied to the Fourier transformed angular spectrum map of the OCM complex-field map in such a way to maximize the image sharpness metric defined by the summation of intensity squared in the image. This algorithm has a major limitation in the case of high-resolution imaging. The Fourier transformed image contains aberrations for both illumination and collection beam paths. Therefore, applying correction to this image cannot deal with the aberration of illumination separately from that in the collection beam path. This may not be a significant issue in the case of low numerical-aperture (NA) imaging, but in high NA imaging the aberration in the illumination beam path causes significant broadening of the illumination PSF.

To elucidate this point, we eliminated both input and output aberrations in the measured time-gated reflection matrix. We then added defocus aberration by $-10\ \mu\text{m}$ to both input and output paths. Figure R4a shows the resulting angular scanned coherently accumulated image of a resolution target before the application of aberration correction, which is equivalent to conventional OCM image. We then scanned the coefficient of the fourth-order Zernike polynomial applied to the Fourier transformed complex-field map of the OCM image. Figure R4c shows the image metric with the scan of the computational refocusing distance using the same method as the reference papers. Two representative image metrics, summation of intensity square and Shannon entropy, were estimated, but there were no perceivable peaks at $10\ \mu\text{m}$. And the computational refocusing back to $10\ \mu\text{m}$ didn't show a clear image (Fig. R4b). This is because the aberration in the illumination path was not dealt with in this reference work. On the other hand, AO-SASM found the aberration maps for both illumination and collection paths (Figs. R4e and f), which correspond to $10\ \mu\text{m}$ defocus, and showed clear object image (Fig. R4d). The main difference of AO-SASM from conventional pinhole-gated OCM is that AO-SASM collects signals at both the point of illumination and the surrounding area. This ultimately enables us to deal with the input and output aberrations separately, thereby enabling high-order aberration correction for high-NA imaging.

Figure R4. Comparison of aberration correction performance with software-based approach using

optical coherence microscopy for high-NA imaging. **a**, Amplitude map numerically defocused by -10 μm for OCM image of a Siemens star pattern. Scale bar: 5 μm . **b**, Amplitude map after applying 10 μm refocusing to the angular spectrum of **a**. **c**, Variation of image metric with the scanning of the coefficient of the Zernike polynomial corresponding to defocusing. **d**, AO-SASM image for the same Siemens star pattern in **a** after applying our algorithm. **e** and **f**, aberration maps for input (**e**) and output (**f**) paths. Color bars: phase in radians.

 To clarify the steps where the input and output aberrations are separately addressed, we added the following sentence to the Methods section.

“The applications of the correction maps $\theta_i^{(1)}(\vec{k}^i)$ to each \vec{k}^i and $\theta_o^{(1)}(\vec{k}^o)$ to each \vec{k}^o form a separate treatment of input and output aberrations.”

In Fig. R9 (which is also Fig. S9), and the associated text, the authors state that the approximate scattering mean free paths were around 50 to 100 μm . If the scattering length $l_s = 50 \mu\text{m}$, then at a depth of 200 μm this is only 4 scattering mean free paths, and the stated “attenuation of single scattering signal by a factor of 10^2 - 10^4 times at the depth of 200 μm ” seems high. Is this for double pass detection, and is it referring to attenuation of the amplitude or power of the single scattering signal? Also, for the sake of clarity it would help to separately mention the impact of multiple scattering vs. aberrations. And so I recommend separating the last part of the sentence “and the sample-induced aberrations further attenuate single scattering intensity” into a separate sentence that separately mentions the impact of aberrations.

As the reviewer conjectured, the attenuation of single scattering signal by a factor of 10^2 - 10^4 accounts for the power attenuation in the double-pass detection. The power attenuation factor is given by e^{-2z/l_s} , where factor 2 in the exponent accounts for the roundtrip. To clarify, we revised the sentence as follows.

“Considering the double-pass detection geometry, this corresponds to the attenuation of single scattering intensity by a factor of 10^2 - 10^4 times at the depth of 200 μm in the case of normal-illumination complex-field map.”

We appreciate the reviewer’s suggestion to separate out the sentence describing the impact of aberrations. Following the reviewer’s suggestion, we added the following sentences with additional explanation.

“The sample-induced aberrations attenuate single scattering intensity by an additional factor of 10~100 in the image formation step. The constructive superposition of multiple angular waves is compromised due to the angle-dependent phase retardations induced by the sample.”

REVIEWERS' COMMENTS:

Reviewer #3 (Remarks to the Author):

The authors thoroughly addressed all the second round review comments. I enthusiastically recommend publication in Nature Communications.